# Amortized Sampling with Transferable Normalizing Flows

**Charlie B. Tan**[*][1]       **Majdi Hassan**[*][2,3]       **Leon Klein**[4]

**Saifuddin Syed**[1]       **Dominique Beaini**[2,3,5]       **Michael M. Bronstein**[1,6]

**Alexander Tong**[†][2,3,6]       **Kirill Neklyudov**[†][2,3,7]

[1]University of Oxford    [2]Université de Montréal    [3]Mila - Quebec AI Institute

[4]Freie Universität Berlin    [5]Valence Labs    [6]AITHYRA    [7]Institut Courtois

## Abstract

Efficient equilibrium sampling of molecular conformations remains a core challenge in computational chemistry and statistical inference. Classical approaches such as molecular dynamics or Markov chain Monte Carlo inherently lack *amortization*; the computational cost of sampling must be paid in full for each system of interest. The widespread success of generative models has inspired interest towards overcoming this limitation through learning sampling algorithms. Despite performing competitively with conventional methods when trained on a single system, learned samplers have so far demonstrated limited ability to transfer across systems. We demonstrate that deep learning enables the design of scalable and transferable samplers by introducing PROSE, a 285 million parameter all-atom *transferable* normalizing flow trained on a corpus of peptide molecular dynamics trajectories up to 8 residues in length. PROSE draws zero-shot uncorrelated proposal samples for arbitrary peptide systems, achieving the previously intractable transferability across sequence length, whilst retaining the efficient likelihood evaluation of normalizing flows. Through extensive empirical evaluation we demonstrate the efficacy of PROSE as a proposal for a variety of sampling algorithms, finding a simple importance sampling-based fine-tuning procedure to achieve competitive performance to established methods such as sequential Monte Carlo. We open-source the PROSE codebase, model weights, and training dataset, to further stimulate research into amortized sampling methods and objectives.

## 1 Introduction

Accurately sampling molecular configurations from the Boltzmann distribution is a fundamental problem in statistical physics with profound implications for understanding biological and chemical systems. Key applications include protein folding [Noé et al., 2009, Lindorff-Larsen et al., 2011], protein–ligand binding [Buch et al., 2011], and crystal structure prediction [Köhler et al., 2023]; processes that underpin advances in drug discovery and material science.

---

[*]Equal contribution. [†]Equal advising.
Correspondence to: `charlie.tan@exeter.ox.ac.uk` and `majdi.hassan@mila.quebec`

39th Conference on Neural Information Processing Systems (NeurIPS 2025).

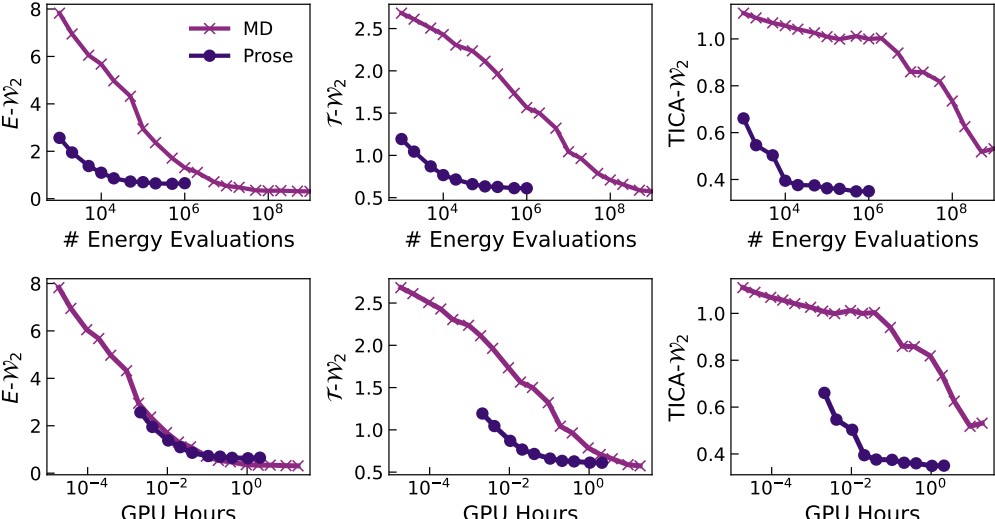

Figure 1: **PROSE exceeds the quantitative performance of molecular dynamics on *unseen* peptide systems.** Wasserstein-2 distances on energy, dihedral torus, and TICA projection with respect to reference molecular dynamics (5 μs), for a (1 μs) molecular dynamics baseline and PROSE (with SNIS), at a range of energy evaluation (above) and GPU walltime budgets (below). Each value represents the mean over 30 unseen tetrapeptide systems. PROSE outperforms the baseline with respect to energy evaluations on all metrics. Whilst comparable on $E$-$\mathcal{W}_2$ for a given time budget, the baseline is significantly inferior on the $\mathcal{T}$-$\mathcal{W}_2$ and TICA-$\mathcal{W}_2$ macrostructure metrics, highlighting long simulation periods were required to traverse the separated metastable states.

Conventional approaches such as Markov chain Monte Carlo (MCMC) [Liu, 2001] and, in particular, Molecular Dynamics (MD) [Leimkuhler and Matthews, 2015] seek to tackle this problem by proposing a general solution, which, however, has practical limitations due to its Markov nature. To accurately integrate the corresponding Hamiltonian dynamics, MD has to be simulated with a fine time-discretization (on the order of femtoseconds), which produces highly correlated samples and prevents efficient exploration of the modes of the Boltzmann density. Although running multiple chains from different initializations is possible, every chain has to be simulated for a long time to ensure proper mixing, which cannot be efficiently parallelized. Finally, the entire simulation has to be started *from scratch* for a new system, which bottlenecks the speed of ab initio studies.

Deep learning-based samplers abandon the Markov chain approach to drawing samples and shift the computational burden to a one-time training phase, enabling fast and inexpensive inference compared to MCMC. In the most challenging scenario, these methods consider having access only to the unnormalized density function (analogous to MC methods) [Vargas et al., 2023, Akhound-Sadegh et al., 2024]. Boltzmann generators (BGs) [Noé et al., 2019] consider a more practical scenario where, in addition to the unnormalized density, a dataset of MD trajectory is available, which does not necessarily match the target density. To mitigate the error introduced by imperfections of the model and training data, BGs train likelihood-based models and perform self-normalized importance sampling (SNIS) [Liu, 2001] at inference time. The availability of training data coupled with SNIS have enabled BGs to generalize across dipeptide systems [Klein and Noe, 2024], but they have not yet been able to generalize across larger and more diverse systems of scientific interest.

In this work, we introduce PROSE, a large-scale normalizing flow which demonstrates unprecedented ability to transfer to previously unseen systems of varying amino acid composition, sequence length, and temperatures, outperforming MD for the same computational budget (see Fig. 1). Our approach is strikingly simple and scalable, which elucidates the potential of the deep learning-based samplers for sampling applications. In particular, we outline the following series of contributions:

- We introduce ManyPeptidesMD: a novel dataset of molecular dynamics trajectories for peptide systems between 2 and 8 residues. The training dataset consists of 21,700 peptide sequences simulated for 200 ns each, giving a total of 4.3 ms of simulation.

- Building on the recently proposed TarFlow [Zhai et al., 2024], we propose architectural modifications, which allow for better modeling of peptide systems, system-transferable conditioning, and generation of peptide sequences of varying length.

- We study the use of PROSE as a proposal distribution for different Monte Carlo algorithms, finding the learned proposal to be sufficiently powerful for accurate sampling with standard SNIS, which does not require tuning of sampling parameters. Furthermore, resampled generations can be used for efficient fine-tuning of PROSE on previously unseen systems.

- Finally, we empirically demonstrate that PROSE achieves state-of-the-art performance when sampling from the equilibrium distribution on previously unseen peptide systems of length up to 8 residues surpassing the continuous normalizing flow-based transferable Boltzmann generator [Klein and Noe, 2024] whilst generating proposals $4 \cdot 10^3$ times faster.

- We open source our codebase `https://github.com/transferable-samplers/transferable-samplers`, ManyPeptidesMD dataset `https://huggingface.co/datasets/transferable-samplers/many-peptides-md` and model weights `https://huggingface.co/transferable-samplers/model-weights`.

## 2 Background

### 2.1 Normalizing flows

The fundamental challenge of probabilistic modeling is designing a density model from which samples can be efficiently generated. Normalizing flows [Rezende and Mohamed, 2015] approach this challenge by defining a diffeomorphism; a differentiable invertible function with a differentiable inverse. Namely, given a simple prior density $q_z(z)$ and a parameterized flow (diffeomorphism) $f_\theta(x)$, one can define the push-forward distribution as the map of samples from the prior distribution $z \sim q_z(z)$ via the inverse flow $x = f_\theta^{-1}(z) \sim q_\theta(x)$ with learnable parameters $\theta$. The density of the push-forward distribution can then be computed via the change-of-variables formula

$$q_\theta(x) = \int dz \, q_z(z) \delta(x - f_\theta^{-1}(z)) = q_z(f_\theta(x)) \left| \frac{\partial f_\theta(x)}{\partial x} \right|, \qquad (1)$$

where $|\partial f_\theta(x)/\partial x|$ is the Jacobian determinant of the map $f_\theta$. In practice, one has to be able to efficiently evaluate $f_\theta^{-1}(z)$ for sample generation, and $|\partial f_\theta(x)/\partial x|$ for likelihood evaluation.

Autoregressive normalizing flows [Kingma et al., 2016, Papamakarios et al., 2017, Zhai et al., 2024] define a family of invertible maps with tractable Jacobian as a sequence of composed transformations $f_\theta = f_\tau \circ \ldots \circ f_0$, where each transformation $z_{t+1} = f_t(z_t), z_0 = x$ is defined autoregressively. In the case of the TarFlow [Zhai et al., 2024], this is an autoregressive affine update defined over blocks of latent variable corresponding to image patches $z_t[i] \in \mathbb{R}^D$. That is, the $i$-th latent block is

$$z_{t+1}[i] = \begin{cases} z_t[i], & i = 0, \\ (z_t[i] - \mu_t(z_t[:i])[i]) \odot \exp(-\alpha_t(z_t[:i])[i]), & i \in [1, N-1], \end{cases} \qquad (2)$$

where we adopt slicing notation denoting the $i$-th block as $z[i]$ and blocks up to the $i$-th (exclusive) as $z[:i]$. Notably, the autoregressive structure allows for efficient evaluation of the Jacobian determinant due to its lower-triangular structure $\log |\partial f_t(z_t)/\partial z_t| = -\sum_{i=1}^{N-1} \sum_{j=0}^{D-1} \alpha_t(z_t[:i])[i]_j$. Furthermore, the autoregressive affine updates are invertible with inverse $z_t = f_t^{-1}(z_{t+1})$ given by

$$z_t[i] = \begin{cases} z_{t+1}[i], & i = 0, \\ z_{t+1}[i] \odot \exp(\alpha_t(z_t[:i])[i]) + \mu_t(z_t[:i])[i], & i \in [1, N-1]. \end{cases} \qquad (3)$$

However, clearly, such transformations leave the leading dimension $z_t[0]$ untouched, hence must be interleaved with permutations $\pi_t$ over the dimensions $f_\theta = \pi_\tau \circ f_\tau \circ \ldots \circ \pi_0 \circ f_0$. For example, Zhai et al. [2024] use simple inversions of the latent block sequence for all $\pi_t$ across the entire model.

### 2.2 Boltzmann generators

Despite normalizing flows allowing various forms of training supervision, such as variational inference for unnormalized densities [Rezende and Mohamed, 2015], or maximum likelihood from empirical distributions [Kingma and Dhariwal, 2018], errors present in the parameterized distribution prevents accurate evaluation within scientific applications requiring high precision, e.g. free energy estimation.

*Boltzmann generators* address specifically this challenge by performing self-normalized importance sampling (SNIS) at inference time. Namely, to evaluate the expectation of a statistic $\varphi(x)$ w.r.t. the target Boltzmann density $p(x)$ one can use the following consistent Monte Carlo estimator

$$\mathbb{E}_{p(x)}\varphi(x) \approx \sum_{i=1}^{n} \frac{w_i}{\sum_{j=1}^{n} w_j}\varphi(x_i)\,, \quad w_i = \frac{p(x_i)}{q_\theta(x_i)}\,, \quad x_i \sim q_\theta(x)\,, \tag{4}$$

where $q_\theta(x)$ is the density of the learned normalizing flow. The SNIS estimator converges, for $n \to \infty$, to the true value $\mathbb{E}_{p(x)}\varphi(x)$. Furthermore, Tan et al. [2025] extended the Boltzmann generator framework to more general Monte Carlo algorithms, in particular a continuous-time formulation of Sequential Monte Carlo [Jarzynski, 1997, Albergo and Vanden-Eijnden, 2025].

Transferable Boltzmann generator (TBG) [Klein and Noe, 2024] made a first attempt of learning a sampler that generalizes across target densities corresponding to different peptide systems. TBG parametrizes the proposal distribution as a continuous normalizing flow (CNF) [Chen et al., 2018b], where the vector field is defined by an equivariant graph neural network [Satorras et al., 2021, Klein et al., 2023c]. Crucial to the method is the system-dependent conditioning of $N$ atoms

$$h[i] = [A_i, R_i, P_i]\,, \tag{5}$$

where atom type $A_i$, residue type $R_i$, and residue position $P_i$ are each encoded as one-hot vectors. Training on a set of MD trajectories for dipeptide systems [Klein et al., 2023b] with this system-conditional encoding enables TBG to generate a proposal for previously unseen dipeptides.

However, despite successful generalization across dipeptides, the TBG architecture introduces significant bottlenecks for inference and fine-tuning. Indeed, the learned CNF requires accurate integration of the vector field and computationally expensive evaluation of its divergence for evaluating the learned density model. For instance, the implementation of [Klein and Noe, 2024] requires 4 GPU-days to produce $3 \times 10^4$ samples with their corresponding proposal likelihoods for a single dipeptide system. Furthermore, the expensive evaluation of density makes it infeasible to train or finetune TBG via the reverse KL-divergence or create a replay buffer of a substantial size.

## 3 Scalable transferable normalizing flows as Boltzmann generators

### 3.1 Architecture of PROSE

PROSE builds on the TarFlow architecture [Zhai et al., 2024], which parametrizes a sequence of autoregressive affine transformations via blocks of transformer layers. The expressivity and favorable scalability of the transformer layers enables TarFlow to effectively model high dimensional data, whilst the affine autoregressive flow parameterization ensure fast and accurate energy evaluation. With minimal modifications TarFlow is capable of successfully modeling high-dimensional molecular data [Tan et al., 2025]. Here we describe our design choices that make transferability possible.

**Transferability across system dimensions.** We extend TarFlow to support concurrent training on sequences of variable length. Whilst transformers natively support sequences of arbitrary length, special consideration is required within a normalizing flow such as TarFlow that is defined for fixed input and output dimensions. We therefore define appropriate masking to the affine sequence updates and log-determinant aggregation to prevent padding tokens influencing either computation, under arbitrary sequence permutations; further details of the tokenization and masking are provided in Appendix A. We additionally replace the fixed-length learnable position embedding with the more extrapolation-friendly sinusoidal embedding. This design enables PROSE to efficiently train across a distribution of systems $s$ by maximizing the normalized log-likelihood

$$\max_\theta \mathbb{E}_s \frac{1}{d(s)}\mathbb{E}_{x \sim p(x\,|\,s)}\log q_\theta(x) \tag{6}$$

where $d(s)$ is the size of the system $s$ [Klein and Noe, 2024]. This extended architecture allows for parallel processing of data dimensions, enabling transferability and scalability across lengths.

**Adaptive system conditioning.** The standard TarFlow employs simple additive conditioning for class-conditional image generation. Whilst we find this to be sufficient to define a system-transferable normalizing flow, we follow large-scale atomistic transformer architectures in applying conditioning

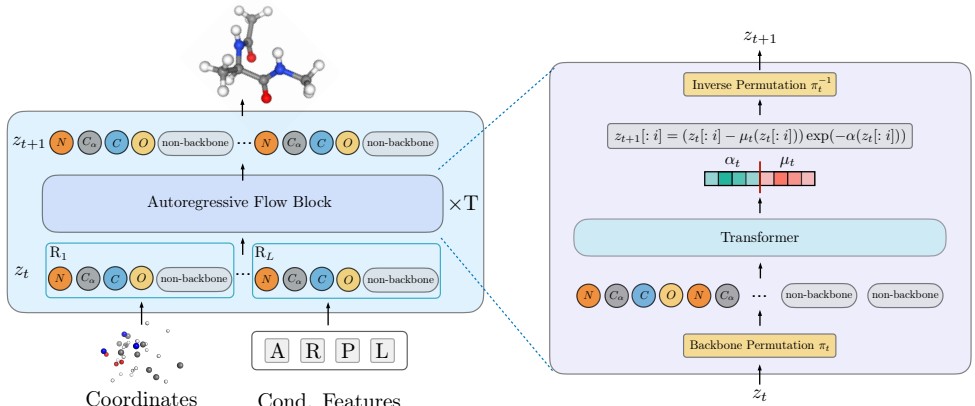

Figure 2: **All-atom block-wise autoregressive normalizing flow based on the TarFlow [Zhai et al., 2024].** Peptides are encoded via atom types $A$, residue types $R$, residue position $P$, and sequence length $L$. Atom positions in 3D Cartesian coordinates define the system state. The embedding of the peptide is applied as conditioning to the coordinates such that PROSE achieves transferability between systems. Within each block the sequence $z_t$ is permuted and passed to a transformer, defining an autoregressive affine update. In the backbone permutation the backbone $[N_i, C_{\alpha,i}, C_i, O_i]_{i=0}^{L-1}$ of all residues (with associated hydrogens) is updated before any sidechains, providing additional diversity to the causal attention for global structure modeling.

through adaptive layer normalization, adaptive scaling, and SwiGLU transition blocks [Abramson et al., 2024, Geffner et al., 2024]. The system conditioning features are constructed from atom types $A$, residue types $R$, residue positions $P$, and sequence lengths $L$. Atom and residue types are embedded using lookup-table embedding layers, whilst sinusoidal embeddings are employed for the naturally ordered sequence position and sequence length. See Appendix A for further details.

**Chemistry-aware sequence permutations.** In the image setting, Zhai et al. [2024] employ only an identity and flip permutation to the sequence of image tokens. Similarly, when applying TarFlow to peptide systems Tan et al. [2025] employ only an identity and flip permutation on the ordering defined per-residue starting with backbone atoms followed by sidechain atoms. Whilst a simple identity and flip may be appropriate for the regular grid of image data, we argue this to be suboptimal for the diversity of geometric interactions present in molecular systems. This motivates our introduction of chemistry-aware sequence permutations, defined to promote effective peptide modeling. We define the *backbone permutation*, such that the backbone atoms $[N_i, C_{\alpha,i}, C_i, O_i]_{i=0}^{L-1}$ (with associated hydrogens) for all residues are located at the start of the sequence, and followed by the sidechains. By processing the coordinates of the backbone atoms at the start of the sequence, the model refines the global structure of the peptide as a contiguous sequence. Crucially, the sidechain positions are subsequently able to causally attend to the full backbone structure, hence enabling local updates to be influenced by global structure. We further employ a *backbone-flip* permutation to provide additional diversity to the autoregressive modeling.

### 3.2 Inference and fine-tuning of PROSE

The practical applicability of learned sampling methods depends significantly on their inference-time throughput, as well as their transferability to unseen systems. In this section, we describe how one can employ PROSE for inference-time importance sampling, importance sampling-based fine-tuning, as well as annealing of the learned proposal to different target temperatures.

**Importance sampling.** At the inference time, one can use PROSE to estimate the expectation of statistics $\varphi(x)$ w.r.t. the target Boltzmann density $p(x)$ via a self-normalized importance sampling (SNIS) estimator. Namely, we consider standard SNIS, discrete-time sequential Monte Carlo (SMC) [Neal, 2001, Doucet et al., 2001], and continuous-time SMC [Jarzynski, 1997, Albergo and Vanden-Eijnden, 2025]. All these estimators are of the form

$$\mathbb{E}_{p(x)}\varphi(x) \approx \sum_{i=1}^{n} \frac{w_i}{\sum_{j=1}^{n} w_j} \varphi(x_i), \quad w_i = \frac{p(x_i)}{q(x_i)}, \quad x_i \sim q(x), \tag{7}$$

where the only difference between them is the proposal density $q(x)$. Note that these estimators can be interpreted as the expectation over the empirical distribution, i.e.

$$\sum_{i=1}^{n} \frac{w_i}{\sum_{j=1}^{n} w_j} \varphi(x_i) = \mathbb{E}_{\tilde{p}(x)} \varphi(x), \quad \tilde{p}(x) = \sum_{i=1}^{n} \frac{w_i}{\sum_{j=1}^{n} w_j} \delta(x - x_i). \tag{8}$$

In practice, we compare the true density $p(x)$ with our generated distribution $\tilde{p}(x)$ instead of measuring statistics $\varphi(x)$. For completeness, we describe all the considered estimators in Appendix D.

**Self-improvement.** For an unseen system $s$ we demonstrate the ability to fine-tune PROSE using a self-improvement strategy. Namely, we iteratively generate the empirical distribution $\tilde{p}(x \,|\, s)$ by resampling the samples from the model $q_\theta(x \,|\, s)$ proportionally to $p(x \,|\, s)$ and use these samples for fine-tuning. Note that this is different from classical fine-tuning as true samples from the target are not available. We update the parameters by maximizing the likelihood on the resampled proposal, i.e.

$$\max_{\theta} \mathbb{E}_{\tilde{p}(x \,|\, s)} \log q_\theta(x \,|\, s), \quad \tilde{p}(x \,|\, s) = \sum_{i=1}^{n} \frac{w_i}{\sum_{j=1}^{n} w_j} \delta(x - x_i), w_i = \texttt{detach}\left( \frac{p(x_i \,|\, s)}{q_\theta(x_i \,|\, s)} \right). \tag{9}$$

This is akin to the energy-based training of Jing et al. [2022], in which samples are proposed by ODE-integration of a diffusion model, resampled, and then used in the score-matching objective.

**Temperature transfer.** Temperature is fundamental to molecular simulation, with significant influence on conformational dynamics and statistic expectations. It is therefore highly desirable that a learned sampler may transfer across temperature without retraining. Formally, we aim to change the temperature $T = 1/\beta$ of the learned density model when generating samples, i.e.

$$\beta \log q_\theta(f_\theta^{-1}(z)) = \beta \log q_z(z) - \beta \log \left| \frac{\partial f_\theta^{-1}(z)}{\partial z} \right|. \tag{10}$$

Note that, for measure-preserving flows $\log |\partial f_\theta^{-1}(z)/\partial z| = 0$, one simply has to change the temperature of the prior distribution (i.e. sample $z \sim q_z(z)^\beta$ instead of $z \sim q_z(z)$) to change the temperature of the density model, which is a standard technique in the normalizing flow literature [Kingma and Dhariwal, 2018, Dibak et al., 2022]. Whilst PROSE is a non-volume preserving flow [Dinh et al., 2017], hence violating this assumption, we found that simply scaling the prior temperature $\beta \log q_z(z)$ results in a suitable proposal for the Boltzmann density with the corresponding temperature.

## 4 Experiments

To establish the performance of PROSE, we first introduce a new dataset of peptide molecular dynamics. We employ this dataset to train PROSE and prior methods, and evaluate using metrics computed against reference molecular dynamics trajectories. We additionally evaluate PROSE as a proposal for a variety of sampling algorithms, and in the temperature-transfer setting.

### 4.1 Molecular dynamics trajectory dataset

We introduce ManyPeptidesMD; a novel dataset of peptide MD trajectories for sequences ranging from 2 to 8 residues in length[1]. Following Klein et al. [2023b] all simulation is performed using `OpenMM` [Eastman et al., 2017] with the Amber14 forcefield [Case et al., 2014]. For training, a total of 21,700 uniformly sampled sequences are simulated for $200 \, \text{ns}$. For evaluation, 30 sequences of length 2, 4, and 8 are randomly sampled such that all amino acids are represented equally, and simulated for $5 \, \mu\text{s}$. Further details on dataset collection and MD configuration provided in Appendix B.

Table 1: Number of sequences used per peptide length for training and evaluation.

| Sequence length | 2 | 3 | 4 | 5 | 6 | 7 | 8 |
|---|---|---|---|---|---|---|---|
| Training | 200 | 1,000 | 1,500 | 2,000 | 3,000 | 4,000 | 10,000 |
| Evaluation | 30 | — | 30 | — | — | — | 30 |

---

[1]Available at `https://huggingface.co/datasets/transferable-samplers/many-peptides-md`

## 4.2 Experimental configuration

We train the first Boltzmann generators transferable across peptide sequence length. We train the PROSE architecture defined in Section 3.1, an unmodified TarFlow [Zhai et al., 2024] as in SBG [Tan et al., 2025], and the equivariant CNF of Klein and Noe [2024], with the improved training recipe of Tan et al. [2025], denoted as ECNF++. All models are trained for $5 \times 10^5$ iterations with batch size 512. Both PROSE and TarFlow are suitably scalable to long sequences and are trained on the full dataset detailed in Section 4.1. However, generating 8 residue sequences with likelihoods for ECNF++ was found to be prohibitively expensive, hence the training data was limited to sequences up to and including length 4. Comprehensive training details are provided in Appendix C.

The primary evaluation metrics are the Wasserstein-2 distance on: (i) the energy distribution $E$-$\mathcal{W}_2$, (ii) the dihedral angle torus distribution $\mathcal{T}$-$\mathcal{W}_2$, (iii) the first 2 TICA component projections TICA-$\mathcal{W}_2$. The energy distribution is highly sensitive to perturbation in bond length and angle, hence $E$-$\mathcal{W}_2$ measures accuracy on fine-grained details. The dihedral angle tori and TICA projection describe macrostructure, hence $\mathcal{T}$-$\mathcal{W}_2$ and TICA-$\mathcal{W}_2$ measure accuracy in terms of metastable state coverage. We additionally report effective sample size (ESS); the variance of the importance weights. For metric definitions and further details on sampling evaluation procedure please refer to Appendix E.

## 4.3 Scale transferability of PROSE

To establish the performance of PROSE as a sampler proposal distribution, we first evaluate the trained flows in the Boltzmann generator setting. Here we generate a set of proposal particles $\{x_i\}_{i=1}^N$, evaluate model likelihoods $q_\theta(x_i)$ and reweight using SNIS as in Eq. (7). In addition to the trained models we benchmark against the following pretrained baselines; (i) the TBG model trained by Klein and Noe [2024], denoted as ECNF (ii) TimeWarp [Klein et al., 2023a] (iii) BioEmu [Lewis et al., 2024] (iv) Unisim [Yu et al., 2025]. For all methods we permit a budget of $10^4$ energy evaluations. For the Boltzmann generator methods (ECNF, ECNF++, TarFlow, PROSE) this corresponds to $10^4$ SNIS particles; for further information on the budget allocation of non-BG methods (TimeWarp, BioEmu, UniSim) see Appendix E. We additional provide results for the unweighted proposal distributions in Appendix F. We note the TimeWarp dataset to lack any sequences containing Proline at the N-Terminal, hence neither TimeWarp nor ECNF were evaluated on such sequences.

Table 2: Quantitative results for baseline methods, and flows with self-normalized importance sampling on peptide systems up to 8 residues. All methods evaluated a budget of $10^4$ energy evaluations. Best values in **bold**. [*]Not evaluated on sequences with N-terminal proline due to absence in training data.

| Sequence length → | 2AA (30 systems) | | | 4AA (30 systems) | | | | 8AA (30 systems) | | | |
|---|---|---|---|---|---|---|---|---|---|---|---|
| Model ↓ | ESS ↑ | $E$-$\mathcal{W}_2$ ↓ | $\mathcal{T}$-$\mathcal{W}_2$ ↓ | ESS ↑ | $E$-$\mathcal{W}_2$ ↓ | $\mathcal{T}$-$\mathcal{W}_2$ ↓ | TICA-$\mathcal{W}_2$ ↓ | ESS ↑ | $E$-$\mathcal{W}_2$ ↓ | $\mathcal{T}$-$\mathcal{W}_2$ ↓ | TICA-$\mathcal{W}_2$ ↓ |
| TimeWarp[*] | — | 4.532 | 0.842 | — | 7.237 | 2.204 | 0.993 | — | — | — | — |
| BioEmu | — | 45.313 | 1.208 | — | 90.079 | 2.037 | 1.479 | — | 193.873 | 4.638 | 1.601 |
| UniSim | — | $> 10^5$ | 1.289 | — | $> 10^4$ | 2.766 | 1.733 | — | $> 10^3$ | 6.156 | 1.495 |
| ECNF[*] | 0.086 | 0.894 | 0.488 | — | — | — | — | — | — | — | — |
| ECNF++ | 0.024 | 3.470 | 0.302 | 0.008 | 10.032 | 1.121 | 0.572 | — | — | — | — |
| TarFlow | 0.134 | 0.452 | **0.193** | 0.045 | 1.260 | 0.924 | 0.492 | 0.008 | 11.298 | 2.733 | 1.087 |
| PROSE | **0.191** | **0.371** | 0.210 | **0.071** | **0.932** | **0.752** | **0.367** | **0.011** | **10.038** | **2.456** | **0.988** |

We present metrics for PROSE and baseline methods in Table 2. PROSE achieves the strongest performance on all metrics aside from dipeptide $\mathcal{T}$-$\mathcal{W}_2$, where it is marginally outperformed by TarFlow, confirming it to be a strong SNIS proposal for peptide systems of varying sequence length. ECNF++ performs very poorly on $E$-$\mathcal{W}_2$ on both dipeptides and tetrapeptides, seemingly unable to learn an effective vector field when trained on tetrapeptides. TimeWarp is the strongest non-Boltzmann generator baseline, with both BioEmu and UniSim attaining high values of $E$-$\mathcal{W}_2$. However, we note the training data for these pretrained models does not correspond exactly with our evaluation data and hence they are not directly comparable. Fig. 1 further confirms the success of PROSE with SNIS as an amortized sampler, surpassing the performance of a baseline MD trajectory on the critical $\mathcal{T}$-$\mathcal{W}_2$ and TICA-$\mathcal{W}_2$ describing metastable state coverage w.r.t. both energy evaluations and GPU walltime. We additionally present qualitative results on the unseen octapeptide `DGVAHALS` in Fig. 3, demonstrating the unprecedented scalability of PROSE; further results are provided in Appendix F.

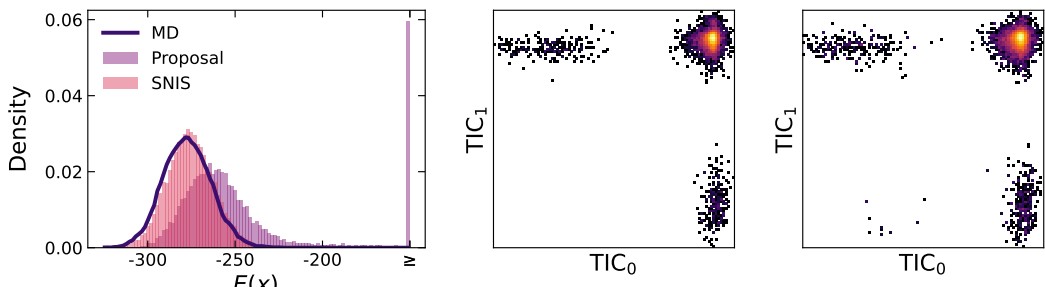

Figure 3: **PROSE accurately samples from the Boltzmann distributions of unseen octapepitde system.** Empirical results for sampling from `DGVAHALS` peptide system, not present in training data. Energy histogram (left) for reference MD data, PROSE proposal and PROSE reweighted using SNIS, demonstrate fine-grained detail accuracy. TICA plots for MD (center) and SNIS-reweighted PROSE (right) illustrate mode coverage.

## 4.4 Architecture ablation study

We proceed to ablate the architectural variations applied in PROSE, as described in Section 3.1; (i) the adaptive system conditioning blocks in the transformer layers (ii) the backbone permutations interleaved into our permutation sequence. We train ablation models using an identical training configuration to that described in Section 4.2. We present metrics for these modifications in Table 3. We observe a significant improvement in all metrics across scales of peptide sequence length, confirming the efficacy of these modifications for atomistic modeling, notably the backbone permutations which introduce negligible runtime complexity over the standard TarFlow architecture.

Table 3: Ablation results for PROSE architecture components. SNIS performed with $2 \times 10^5$ energy evaluations.

| Sequence length → | 2AA (30 systems) | | | 4AA (30 systems) | | | | 8AA (30 systems) | | | |
|---|---|---|---|---|---|---|---|---|---|---|---|
| Model ↓ | ESS ↑ | $E\text{-}\mathcal{W}_2$ ↓ | $\mathcal{T}\text{-}\mathcal{W}_2$ ↓ | ESS ↑ | $E\text{-}\mathcal{W}_2$ ↓ | $\mathcal{T}\text{-}\mathcal{W}_2$ ↓ | TICA-$\mathcal{W}_2$ ↓ | ESS ↑ | $E\text{-}\mathcal{W}_2$ ↓ | $\mathcal{T}\text{-}\mathcal{W}_2$ ↓ | TICA-$\mathcal{W}_2$ ↓ |
| PROSE | **0.191** | **0.282** | 0.177 | **0.071** | **0.646** | **0.607** | **0.349** | **0.011** | **9.360** | **2.019** | **0.960** |
| w/o Backbone-first | 0.170 | 0.295 | **0.152** | 0.051 | 0.816 | 0.697 | 0.421 | 0.009 | 10.261 | 2.275 | 1.044 |
| w/o Transition | 0.152 | 0.322 | 0.282 | 0.054 | 0.880 | 0.691 | 0.384 | 0.009 | 11.384 | 2.209 | 1.012 |

## 4.5 Sampling algorithms

Having established the unmatched performance of PROSE in the standard Boltzmann generator framework, we now consider alternative sampling algorithms made tractable by its efficient likelihood. We evaluate SNIS, SMC in continuous time, SMC in discrete time, and the simple instantiation of self-improvement defined in Section 3.2. All methods are permitted a budget of $10^6$ energy evaluations, further details on method configurations are provided in Appendix E. Metric results are presented in Table 4. These results reveal the surprising result that, given a suitably strong proposal distribution, SNIS is competitive with both SMC variants, despite requiring no tuning. While SMC discrete achieves the best value of $E\text{-}\mathcal{W}_2$ on octapeptides, both SMC variants introduce a notable deterioration of macrostructure metrics at this scale when compared to SNIS. Furthermore, the performance of SNIS with self-improvement at improving the tetra- and octapeptide $E\text{-}\mathcal{W}_2$ provides strong evidence in favor of proposal fine-tuning within sampling methods, as an alternative to resource allocation solely on annealing-based methods. We provide results for the unseen `RLMM` system in Fig. 4, illustrating the superior mode coverage of PROSE with SNIS over a MD baseline given an allocation of $10^6$ energy evaluations, further evidence of successful amortized sampling with PROSE.

Table 4: Results for samplers using PROSE as proposal. Methods provided with budget of $10^6$ energy evaluations. Best values **bolded**. [*]Not evaluated on sequences with N-terminal proline due to absence in training data.

| Sequence length → | 2AA (30 systems) | | | 4AA (30 systems) | | | | 8AA (30 systems) | | | |
|---|---|---|---|---|---|---|---|---|---|---|---|
| Algorithm ↓ | ESS ↑ | $E\text{-}\mathcal{W}_2$ ↓ | $\mathcal{T}\text{-}\mathcal{W}_2$ ↓ | ESS ↑ | $E\text{-}\mathcal{W}_2$ ↓ | $\mathcal{T}\text{-}\mathcal{W}_2$ ↓ | TICA-$\mathcal{W}_2$ ↓ | ESS ↑ | $E\text{-}\mathcal{W}_2$ ↓ | $\mathcal{T}\text{-}\mathcal{W}_2$ ↓ | TICA-$\mathcal{W}_2$ ↓ |
| Timewarp[*] | — | 2.551 | 0.580 | — | 4.125 | 1.600 | 0.813 | — | — | — | — |
| SNIS | 0.190 | 0.271 | 0.165 | 0.070 | 0.665 | 0.613 | 0.349 | 0.011 | 9.386 | **2.012** | **0.964** |
| SMC Continuous | — | 0.318 | 0.177 | — | 0.764 | 0.688 | **0.338** | — | 10.563 | 2.642 | 1.049 |
| SMC Discrete | — | **0.249** | **0.147** | — | 0.653 | 0.721 | 0.400 | — | **7.672** | 2.524 | 1.086 |
| Self-Improve + SNIS | 0.189 | 0.265 | 0.171 | 0.070 | **0.568** | **0.611** | 0.345 | 0.011 | 8.886 | 2.071 | 0.966 |

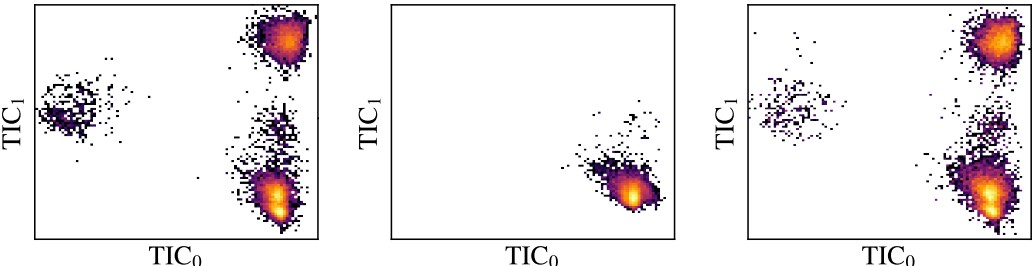

Figure 4: **By drawing *uncorrelated* proposal samples, PROSE achieves greater metastable state coverage than molecular dynamics for the same number of energy evaluations.** TICA projection plots for unseen tetrapeptide system (`RLMM`). After $5 \cdot 10^9$ energy evaluations the reference molecular dynamics (left) has traversed four distinct metastable states, taken to be ground truth. However, with an energy evaluation budget of $10^6$ molecular dynamics explores only a single metastable state (center), highlighting the limitations of simulation-based sampling methods for mode exploration. PROSE with SNIS (right) samples all 4 states given the same budget of energy evaluations, indicating successful amortization of the mode exploration problem.

## 4.6 Inference-time temperature transfer

We lastly evaluate the scaled prior (SP) technique for inference-time temperature transfer introduced in Section 3.2. We collect additional $1\,\mu s$ MD trajectories for the `RLMM` unseen tetrapeptide at temperatures defined by geometric series between the base model temperature of $310\,K$, and $800\,K$. We then perform SNIS using $2 \times 10^5$ energy evaluations from PROSE, both naively and with the scaled prior inference method. Results are presented in Fig. 5, with scaled prior universally outperforming naive SNIS. We emphasize that scaled prior *does not* require any fine-tuning and introduces negligible increase in complexity at inference. These results thus demonstrate that PROSE is transferable not only in system, but also in temperature, opening a variety of avenues of further exploration.

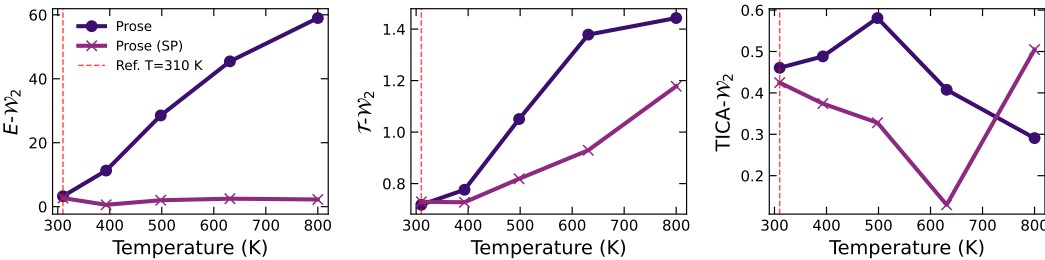

Figure 5: **Scaled prior greatly improves the ability of PROSE to accurately reweight to arbitrary temperatures.** Metrics for PROSE on `RLMM` unseen tetrapeptide, targeting temperatures up to $800\,K$. Naively applying SNIS to the target temperature leads to a rapid degradation in energy distribution, and to a lesser extent the dihedral angle distribution. Applying prior scaling (PROSE SP) leads to a significant improvement in energy distribution at high temperatures and moderate improvement in dihedral angles. Notably, the TICA distribution *improves* at higher temperatures irrespective of scaled prior usage, although scaled prior remains more effective.

## 5 Related Work

**Normalizing flows.** Normalizing flows [Rezende and Mohamed, 2015, Dinh et al., 2017, Kingma and Dhariwal, 2018, Durkan et al., 2019] fell from favor as general-purpose generative models as generative adversarial networks (GANs) [Goodfellow et al., 2014], diffusion models [Ho et al., 2020, Song et al., 2021], and continuous normalizing flows [Chen et al., 2018a, Liu, 2022, Albergo and Vanden-Eijnden, 2023], demonstrated superior empirical generative quality. However, they have still found relevance in scientific applications where efficient likelihood calculations are necessary. Furthermore, the recent introduction of Transformer-based normalizing flows [Zhai et al., 2024, Kolesnikov et al., 2024] has enabled previously intractable data distributions to be modeled whilst retaining efficient likelihood evaluation, and brought renewed research attention to this area.

**Boltzmann generators.** Boltzmann generators are machine learning-based samplers that train likelihood-based models and employ inference-time SNIS to achieve consistent sampling of the target density [Noé et al., 2019]. A major limitation of standard Boltzmann generators is the need to train the proposal model on a dataset of true density samples, motivating methods that transfer between molecular systems. Accordingly, Jing et al. [2022] propose a transferable Boltzmann generator operating on the torsion angles of small molecules, and Klein and Noe [2024] develop a Boltzmann generator operating on Cartesian coordinates that transfers between dipeptide systems. However, the scalability of such methods has remained limited due to the difficulty of designing expressive generative models that possess efficient and accurate likelihood evaluations. In particular the use of continuous normalizing flows implies a large cost to proposal likelihoods due to the need to integrate the vector field divergence [Grathwohl et al., 2019]. Schopmans and Friederich [2025] replace the single-step SNIS of standard Boltzmann generators with a temperature-annealing sequence of normalizing flows, performing SNIS with each flow to sample from a given target distribution.

**Approaches to machine learning-based sampling.** The widespread empirical success of generative modeling has inspired many approaches to machine learning-based sampling. Boltzmann emulators, like Boltzmann generators, seek uncorrelated sampling of the target density, but forgo efficient likelihood evaluation in favor of scalable generative modeling on large pre-collected datasets [Abdin and Kim, 2024, Wayment-Steele et al., 2024, Lewis et al., 2025]. In this case the lack of efficient likelihood evaluation precludes the use of Monte Carlo estimators such as SNIS. Diffusion samplers, propose novel objectives for training diffusion models in the absence of an empirical data distribution, both simulation-based [Berner et al., 2024, Vargas et al., 2023, Richter et al., 2024, Zhang and Chen, 2022, Vargas et al., 2024] and simulation-free [Akhound-Sadegh et al., 2024, Huang et al., 2021, De Bortoli et al., 2024]. Notably, Havens et al. [2025] develop a diffusion sampler that is transferable across small molecules. Time coarseners are another family of model, in which ML is used to predict large time transitions for simulation [Schreiner et al., 2023, Fu et al., 2023, Klein et al., 2023b, Daigavane et al., 2024, Yu et al., 2025], whereas methods like MDGen apply generative modeling to both the spatial and temporal dimensions of MD data [Jing et al., 2024]. Lastly, several works integrate normalizing flows with classical Monte Carlo methods [Albergo et al., 2019, Arbel et al., 2021, Gabrié et al., 2021, Matthews et al., 2022, Midgley et al., 2023b, Hagemann et al., 2023].

## 6  Conclusion

We develop PROSE, demonstrating that deep learning-based samplers can efficiently transfer to previously unseen systems at unprecedented scale. PROSE outperforms learned baseline methods, as well as molecular dynamics, at a variety of energy evaluation and walltime budgets. Notably, PROSE demonstrates state-of-the-art performance whilst retaining many simple design choices; thus, leaving many directions for further development. The competitive performance of SNIS compared to SMC invites further investigation into the merits of annealing-based samplers given a proposal with good coverage of the target density. Naturally, annealing-based samplers have been enhanced beyond the simple instantiations we explore; careful tuning of SMC may yield further improvements [Syed et al., 2024]. We lastly note the self-improvement strategy discussed to not be restricted to SNIS, the integration of advanced Monte Carlo methods presents an avenue for future work.

**Limitations.** Whilst conventional Monte Carlo algorithms make no assumption on the target density, transferable learned samplers, including PROSE, rely on the assumption that the system belongs to a structured space of energy functions, in our case the chemical space of peptides. To achieve greater practical relevance it will be necessary to consider a more diverse chemical space, such as the recent OMol25 [Levine et al., 2025] dataset. We lastly comment that, despite the scaled prior method demonstrating surprising abilities to sample from the higher temperatures, we believe that precise transfer to lower temperatures would require further algorithmic development.

## Acknowledgments

This research is partially supported by the EP- SRC Turing AI World-Leading Research Fellowship No. EP/X040062/1 and EPSRC AI Hub No. EP/Y028872/1. The authors acknowledge funding from UNIQUE, CIFAR, NSERC, Intel, and Samsung. The research was enabled in part by computational resources provided by the Digital Research Alliance of Canada (`https://alliancecan.ca`), Mila (`https://mila.quebec`), and NVIDIA. The authors additionally thank HuggingFace for hosting the ManyPeptidesMD dataset. KN was supported by IVADO and Institut Courtois.

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

# Appendix

# A PROSE architecture Details

## A.1 Tokenization and masking

As in SBG [Tan et al., 2025], the token sequence is constructed using a single atom per token $x[i] \in \mathbb{R}^3$. As different peptide sequences contain varying numbers of atoms, we zero-pad atom sequences to a fixed maximum sequence length and introduce a padding mask $m[i] \in 0, 1$ indicating valid ($m[i] = 1$) or padded ($m[i] = 0$) tokens. In the context of a causal transformer, the implementation is greatly simplified by ensuring all padding tokens are placed at the end of the token sequence, irrespective of the permutations applied. We may therefore state the following block update rules, as only a minor adaptation of Eq. (2)

$$z_{t+1}[i] = \begin{cases} z_t[i], & i = 0, \\ (z_t[i] - \mu_t(z_t[: i])[i]) \odot \exp(-\alpha_t(z_t[: i])[i]), & m[i] = 1, \\ [0, 0, 0], & m[i] = 0, \end{cases} \tag{11}$$

with log-determinant of Jacobian given by

$$\log\left|\frac{\partial f_t(z_t)}{\partial z_t}\right| = -\sum_{i=1}^{N-1} \sum_{j=0}^{D-1} m[i] \cdot \alpha_t(z_t[: i])[i]_j. \tag{12}$$

The inverse transformation is correspondingly a minor adaptation of Eq. (3)

$$z_t[i] = \begin{cases} z_{t+1}[i], & i = 0, \\ z_{t+1}[i] \odot \exp(\alpha_t(z_t[: i])[i]) + \mu_t(z_t[: i])[i], & m[i] = 1, \\ [0, 0, 0], & m[i] = 0. \end{cases} \tag{13}$$

## A.2 Permutations

All permutations are defined from the N-terminal to the C-terminal. In the residue-by-residue permutation the atoms are ordered such that each residue forms a contiguous sequence, with sidechain atoms immediately following the corresponding backbone atoms. In the backbone-first permutation the entire sequence of backbone atoms is placed at the start of the sequence before any sidechain atoms. Where constituent residues possess a branch or ring we also introduce a variant in which the branch ordering is flipped or ring traversal is inverted. The flip permutations are a simple inversion of the permutation, where padding tokens and not moved from their position at the end of the sequence. The specific sequence of permutations employed in PROSE is presented in Table 5.

Table 5: Autoregressive permutation order used across the eight transformation blocks.

| Permutation | Description |
|---|---|
| $\pi_0$ | Backbone first |
| $\pi_1$ | Residue–by–residue (flip) |
| $\pi_2$ | Backbone–first (flip) |
| $\pi_3$ | Residue–by–residue |
| $\pi_4$ | Backbone–first (variant) |
| $\pi_5$ | Residue–by–residue (variant, flip) |
| $\pi_6$ | Backbone–first (variant, flip) |
| $\pi_7$ | Residue–by–residue (variant) |

## A.3 Adaptive layer norm and transition

PROSE integrates the adaptive layer normalization and SwiGLU-based [Shazeer, 2020] transition modules employed by Geffner et al. [2024] into the transformer blocks of the TarFlow architecture [Zhai et al., 2024]. The positions of the latent vector $z_t$ are encoded using a sinusoidal positional encoding, which are added directly to $z_t$. The conditional embedding is used in the adaptive layer normalization and adaptive scale components.

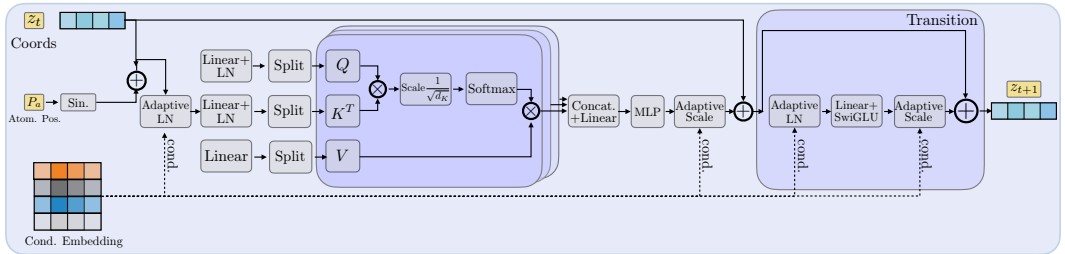

Figure 6: **Adaptive Layer Norm and Transition.** The transformer block is modified to incorporate conditional information using adaptive layer normalization and a transition block. Figure adapted from Geffner et al. [2024].

## B    Dataset

**Sequence sampling.** Training sequences are collected for all peptide lengths by uniformly sampling the 20 standard amino acids. For the 8-residue test data, a sequence of length $30 \cdot 8 = 240$ is constructed by concatenating 12 of each amino acid. This sequence is then randomly permuted and split into peptides of length 8, ensuring that each amino acid is represented uniformly. A similar process is performed for length 4 but was not possible at length 2. In both training and test sets, the N- and C-terminal residues are protonated to form the zwitterionic state of the peptides. Initial structure files (PDB format) are generated using AmberTools' `tleap`.

**Molecular dynamics simulation.** Local energy minimization is performed with the Limited-memory Broyden–Fletcher–Goldfarb–Shanno (L-BGFS) algorithm. Energy minimization is followed by burn-in simulation of length 50 ps, after which samples are collected every 1 ps (train) or 10 ps (test) until the simulation budget is exhausted. Full MD simulation parameters are provided in Table 6.

Table 6: `OpenMM` simulation parameters.

| | |
|---|---|
| Force field | Amber14 |
| Integration time step | 1 fs |
| Friction coefficient | $0.3\,\mathrm{ps}^{-1}$ |
| Temperature | 310 K |
| Nonbonded method | `CutoffNonPeriodic` |
| Nonbonded cutoff | 2 nm |
| Integrator | `LangevinMiddleIntegrator` |

Table 7: Training and evaluation dataset parameters.

| | Train | Test |
|---|---|---|
| Burn-in period | 50 ps | 50 ps |
| Sampling interval | 1 ps | 10 ps |
| Simulation time | 200 ns | 5 μs |

## C    Training configuration

All models are trained for $5 \cdot 10^5$ iterations using a batch size of 512 with the AdamW optimizer [Loshchilov and Hutter, 2018]. We employ a cosine learning rate schedule in which the initial and final learning rates are a reduction of the maximal value by factor of 500, as well as exponential moving average with decay of 0.999. No overfitting was observed hence no early stopping was required. Given the large size of the training trajectories, we subsample to 10 ps per frame. Samples are normalized using values approximating the standard deviation of the 8AA data ($\sigma = 0.35$), or if

absent the 4AA data ($\sigma = 0.28$), noting that a single value must be shared across systems of different dimensionality. An overview of all training configurations is provided in Table 8.

**Continuous Normalizing Flows.** We use the ECNF++ training recipe defined by Tan et al. [2025]; this entails a learning rate of $5 \cdot 10^{-4}$ and weight decay of $1 \cdot 10^{-4}$, with default AdamW hyperparameters of AdamW $\beta_1, \beta_2$ of $(0.9, 0.999)$. In contrast the ECNF of Klein and Noe [2024] was trained without weight decay or exponential weight averaging. The channel width and layer depth of both models is defined in Table 9.

**TarFlows.** Following Zhai et al. [2024] and Tan et al. [2025] we use a learning rate of $1 \cdot 10^{-4}$, weight decay of $4 \cdot 10^{-4}$, and AdamW $\beta_1, \beta_2$ of $(0.9, 0.95)$. Data augmentation is applied as random rotations and Gaussian center of mass augmentation, in which every the entire system conformation is translated by a vector $c \sim \mathcal{N}(0, \sigma_c^2 I_3)$. The $\sigma^2$ value is chosen to match that of the prior, which has a center of mass variance $\sigma_c^2 = \frac{1}{N}$ where $N$ is the number of atoms. Given $N$ is in our case variable for a single model trained on multiple systems, this augmentation is applied-per system before padding is applied. The architecture width and depth is provided in Table 9.

Table 8: Overview of training configurations.

|  | ECNF | ECNF++ | TarFlow / PROSE |
|---|---|---|---|
| Learning Rate | $5 \cdot 10^{-4}$ | $5 \cdot 10^{-4}$ | $1 \cdot 10^{-4}$ |
| Weight Decay | 0.0 | $1 \cdot 10^{-2}$ | $4 \cdot 10^{-4}$ |
| $\beta_1, \beta_2$ | 0.9, 0.999 | 0.9, 0.999 | 0.9, 0.95 |
| EMA Decay | 0.0 | 0.999 | 0.999 |

Table 9: Overview of model scaling parameters. For TarFlow variants depth corresponds to number of parameterized transformations, for ECNF variants this is simply the number of graph neural network layers.

|  | ECNF | ECNF++ | TarFlow | PROSE |
|---|---|---|---|---|
| Channels | 128 | 256 | 384 | 384 |
| Depth | 9 | 9 | 8 | 8 |
| Layers per block | N/A | N/A | 8 | 8 |
| Parameters (M) | 1 | 4 | 115 | 285 |

## C.1 Computational resources

All training experiments are run NVIDIA H100 GPUs using distributed data parallelism. The training throughput for each model is presented in Table 10.

Table 10: Training throughput for models presented in Table 2. We highlight ECNF++ to be trained only on sequences up to length 4, whereas TarFlow and PROSE are trained on sequences up to length 8.

|  | ECNF++ | TarFlow | PROSE |
|---|---|---|---|
| Training iterations / H100 hour | 960 | 1132 | 260 |

## D  Importance sampling variants

### D.1  Self-normalized importance sampling

Self-normalized importance sampling (SNIS) corresponds to the following estimator

$$\mathbb{E}_{p(x)}\varphi(x) \approx \sum_{i=1}^{n} \frac{w_i}{\sum_{j=1}^{n} w_j} \varphi(x_i), \quad w_i = \frac{p(x_i)}{q(x_i)}, \quad x_i \sim q(x), \tag{14}$$

where one uses the learned density model of PROSE $q_\theta(x)$ as $q(x)$.

## D.2 Continuous-time annealed importance sampling

Below, we repeat the derivations from [Jarzynski, 1997, Albergo and Vanden-Eijnden, 2025]. Namely, we consider the continuous family of marginal densities

$$q_t(x) = \frac{1}{Z_t} \exp(-E_t(x)), \quad Z_t = \int dx \ \exp(-E_t(x)). \tag{15}$$

The PDE describing the time-evolution of this density is

$$\frac{\partial q_t(x)}{\partial t} = q_t(x) \left[ -\frac{\partial E_t(x)}{\partial t} + \mathbb{E}_{q_t(x)} \frac{\partial E_t(x)}{\partial t} \right] \tag{16}$$

$$= \pm \left\langle \nabla, q_t(x) \frac{\sigma_t^2}{2} \nabla \log q_t(x) \right\rangle + q_t(x) \left[ -\frac{\partial E_t(x)}{\partial t} + \mathbb{E}_{q_t(x)} \frac{\partial E_t(x)}{\partial t} \right] \tag{17}$$

$$= -\left\langle \nabla, q_t(x) \frac{\sigma_t^2}{2} \nabla \log q_t(x) \right\rangle + \frac{\sigma_t^2}{2} \Delta q_t(x) + q_t(x) \left[ -\frac{\partial E_t(x)}{\partial t} + \mathbb{E}_{q_t(x)} \frac{\partial E_t(x)}{\partial t} \right]. \tag{18}$$

This is a Feynman-Kac PDE which can be simulated [Del Moral, 2013] as the following SDE on the extended space of states $x_t$ and weights $w_t$

$$dx_t = -\frac{\sigma_t^2}{2} \nabla E_t(x) dt + \sigma_t dW_t, \quad x_{t=0} \sim q_0(x)$$

$$d \log w_t = -\frac{\partial E_t(x)}{\partial t} dt, \quad w_{t=0} = 1. \tag{19}$$

The expectation of the statistics $\varphi(x)$ w.r.t. the density $q_T(x)$ then can be estimated using SNIS as follows

$$\mathbb{E}_{q_T(x)} \varphi(x) \approx \sum_{i=1}^{n} \frac{w_T^i}{\sum_{j=1}^{n} w_T^j} \varphi(x_T^i), \tag{20}$$

where $(x_T^i, w_T^i)$ are the solutions of the SDE Eq. (19).

For the inference time of PROSE, we define the continuous family of marginals as

$$q_t(x) \propto \exp\left( \underbrace{(1-t) \log q_\theta(x) + t \log p(x)}_{-E_t(x)} \right), \quad t \in [0, 1], \tag{21}$$

where $q_\theta(x)$ is the learned density of PROSE. Thus, Eq. (19) becomes

$$dx_t = \frac{\sigma_t^2}{2} ((1-t) \nabla \log q_\theta(x) + t \nabla \log p(x)) dt + \sigma_t dW_t, \quad x_{t=0} \sim q_0(x)$$

$$d \log w_t = (\log p(x) - \log q_\theta(x)) dt, \quad w_{t=0} = 1. \tag{22}$$

## D.3 Discrete-time annealed importance sampling

Consider a sequence of marginal densities

$$q_0(x) \propto \exp(-E_0(x)), \ldots, q_K(x) \propto \exp(-E_K(x)). \tag{23}$$

Let's denote by $k_t(x_t \mid x_{t-1})$ the kernel that satisfies the detailed balance w.r.t. $q_t(x_t)$, i.e.

$$q_t(x_{t-1}) k_t(x_t \mid x_{t-1}) = q_t(x_t) k_t(x_{t-1} \mid x_t). \tag{24}$$

Then, one can write importance sampling estimator for the final marginal as

$$\int dx_K \ q_K(x_K) \varphi(x_K) = \int dx_{K-1} dx_K \ k_K(x_{K-1} \mid x_K) q_K(x_K) \varphi(x_K) \tag{25}$$

$$= \int dx_{K-1} dx_K \ k_K(x_K \mid x_{K-1}) q_K(x_{K-1}) \varphi(x_K) \tag{26}$$

$$= \mathbb{E}_{q_{K-1}(x_{K-1}) k_K(x_K \mid x_{K-1})} \frac{q_K(x_{K-1})}{q_{K-1}(x_{K-1})} \varphi(x_K). \tag{27}$$

Clearly, we can repeat the trick but now for $q_{K-1}(x_{K-1})$. Thus, applying this trick recursively to different marginals, we have

$$\int dx_K \, q_K(x_K)\varphi(x_K) = \mathbb{E}_{q_0(x_0)}\mathbb{E}_{x_1,\dots,x_K} \prod_{t=0}^{K-1} \frac{q_{t+1}(x_t)}{q_t(x_t)}\varphi(x_K)\,, \tag{28}$$

$$\text{where } x_1,\dots,x_K \sim k_1(x_1\,|\,x_0)\dots k_{K-1}(x_{K-1}\,|\,x_{K-2})k_K(x_K\,|\,x_{K-1})\,. \tag{29}$$

Thus, we have the following SNIS estimator

$$\int dx \, q_K(x)\varphi(x) \approx \sum_i \frac{w_K^i}{\sum_j w_K^j}\varphi(x_K^i)\,, \tag{30}$$

$$\text{where } x_t^i \sim k_t(x_t\,|\,x_{t-1})\,, \quad t = 1,\dots,K\,, \quad x_0 \sim q_0(x_0) \tag{31}$$

$$\log w_t^i = -E_t(x_{t-1}) + E_{t-1}(x_{t-1}) + \log w_{t-1}^i\,. \tag{32}$$

Note that there is a lot of flexibility for the choice of $k_t(x_t\,|\,x_{t-1})$ because we do not use the densities of the transition kernel for the weights. In particular, the Metropolis-Hastings algorithm with any proposal yields a reversible kernel (satisfies the detailed balance), which result in a consistent final estimator. Furthermore, compared to the continuous-time AIS, discrete-time AIS does not introduce the time-discretization error.

At the inference step of PROSE, we choose

$$q_t(x) \propto \exp\left(\underbrace{\big((1-t)\log q_\theta(x) + t\log p(x)\big)}_{-E_t(x)}\right)\,, \quad t = 0, \frac{1}{K}, \frac{2}{K},\dots,1\,, \tag{33}$$

and Metropolis-Adjusted Langevin Dynamics as the transition kernel $k_t(x_t\,|\,x_{t-1})$ [Roberts and Tweedie, 1996].

## D.4 Sequential Monte Carlo

Sequential Monte Carlo (SMC) [Doucet et al., 2001] can be understood as annealed importance sampling equipped with *adaptive resampling*. As in AIS, particles and weights are propagated in a coupled system. However, SMC tracks the effective sample size (ESS)

$$\text{ESS}(w^1,\dots,w^N) = \frac{\left(\sum_{i=1}^N w^i\right)^2}{\sum_{i=1}^n (w^i)^2}\,, \tag{34}$$

and performs a resampling step whenever ESS falls below a chosen threshold. This is in contrast to AIS, in which resampling only occurs in the final timestep. This adaptive resampling seeks to mitigate weight degeneracy, in which large compute allocations are required in AIS to propogate very low weight particles. While the continuous-time and discrete-time AIS variants are both suitable for SMC, only the continuous-time variant was considered by Tan et al. [2025].

## E Evaluation configuration

### E.1 Proposal sampling and likelihood evaluation

**Equivariant continuous normalizing flows.** Sampling from a continuous normalizing flow (CNF) involves solving the ODE defined by the parameterized vector field $u_t : [0,1] \times \mathbb{R}^{n\times 3} \to \mathbb{R}^{n\times 3}$

$$\frac{dx_t}{dt} = u_t(x_t), \quad x_0 \sim p_0 \tag{35}$$

The corresponding likelihoods can be obtained using the instantaneous change of variables formula

$$\log p_1(x_1) = \log p_0(x_0) - \int_0^1 \nabla \cdot u_t(x_t)dt\,, \tag{36}$$

where $\nabla\cdot$ is the divergence operator. In practice both Eq. (35) and Eq. (36) can be integrated simultaneously with an ordinary differential equation (ODE) solver. We use the Dormand–Prince-5 (dopri5) adaptive solver in all ECNF experiments. Given the $E(3)$ equivariance of the ECNF, samples are generated in both possible global chiralities. Following Klein and Noe [2024] we check for incorrect global sample chirality and flip samples appropriately to match the L-amino acids present in the evaluation data. Unlike Klein and Noe [2024] we do not omit any samples with unresolvable chirality. We additionally apply logit clipping, removing the samples with the 0.2% highest importance weights before resampling Midgley et al. [2023a].

**TarFlow variants.** As discussed in Section 2.1, samples are generated from a normalizing flow simply by applying $f_\theta$ to prior samples $z \sim \mathcal{N}(0, I_{N\times D})$. Model likelihoods are obtained using the change of variables formula (Eq. (1)). Given the lack of translation equivariance in the TarFlow, and the data augmentation applied during training, samples are generated with an approximate scaled $\chi_3$ distribution over centroid norm $||c|| = ||\frac{1}{N}\sum_{i=1}^{N} x_{i,:}|| \sim \sigma\chi_3$. This leads to adverse behavior when resampling with finite samples, hence we apply the center of mass adjustment of [Tan et al., 2025], in which the $\chi_3$ probability density function is divided out of the proposal likelihoods

$$\log p_\theta^c(x) = \log p_\theta(x) - \left[\log\left(\frac{||c||^2}{\sigma^3}\right) + \frac{||c||^2}{2\sigma^2} - \log\left(\sqrt{2}\Gamma\left(\frac{3}{2}\right)\right)\right] \tag{37}$$

where $\Gamma$ is the gamma function. This adjustment seeks to account for the radial component introduced by translation non-equivariance. We additionally apply the same weight clipping threshold as in the ECNF when performing SNIS, or before SMC.

## E.2 Metrics

We report both effective sample size and a variety of Wasserstein-2 distances as evaluation metrics. For the Wasserstein distances a subsample of $10^4$ samples are randomly sampled from the evaluation trajectory as ground truth. Similarly, at most $10^4$ generated samples are employed; if a method has generated more samples a random subset is drawn without replacement.

**Effective sample size.** We compute the effective sample size (ESS) using Kish's formula, normalized by the number of samples generated

$$\text{ESS}\left(\{w_i\}_{i=1}^{N}\right) = \frac{\left(\sum_{i=1}^{N} w_i\right)^2}{N\sum_{i=1}^{N} w_i^2}. \tag{38}$$

**Empirical Wasserstein distance.** We compare generated samples to ground truth data, collected as defined in Appendix B, using empirical Wasserstein-2 distances. Given empirical distributions $\mu = \frac{1}{n}\sum_{i=1}^{n}\delta_{x_i}$ and $\nu = \frac{1}{m}\sum_{j=1}^{m}\delta_{y_j}$, the empirical Wasserstein-2 distance is defined as

$$W_2(\mu, \nu) = \min_{\pi\in\Pi(\mu,\nu)}\sqrt{\sum_{i=1}^{n}\sum_{j=1}^{m}\pi_{ij}\,c(x_i, y_j)^2} \tag{39}$$

where $\Pi(\mu, \nu)$ denotes the set of couplings with marginals $\mu$ and $\nu$, and $c(x, y)^2$ is a defined cost function. Different choices of $c(x, y)^2$ define different measures of dissimilarity. We use the POT [Flamary et al., 2021] linear optimal transport solver to compute the optimal couplings.

**Energy cost.** The energy of a sample $E(x)$ is sensitive to both bonded forces and non-bonded forces. For the energy Wasserstein-2 distance $E\text{-}\mathcal{W}_2$ the cost function is simply

$$c_E(x, y)^2 = |E(x) - E(y)|^2 \tag{40}$$

**Dihedral torus cost.** The $\phi$ and $\psi$ backbone dihedral angles of a peptide conformation encode essential information regarding secondary and tertiary structure. We compare generated and ground truth samples in angle space by defining the dihedral angle vector

$$\text{Dihedrals}(x) = (\phi_1, \psi_1, \phi_2, \psi_2, \ldots, \phi_{L-1}, \psi_{L-1}) \tag{41}$$

where $L$ is the number of residues. Given the torus geometry implied by angle periodicity $\phi_i \in (-\pi, \pi]$, a natural cost function is the minimal signed angle difference

$$c_{\mathcal{T}}(x,y)^2 = \sum_{i=1}^{2L} [(\text{Dihedrals}(x)_i - \text{Dihedrals}(y)_i + \pi) \bmod 2\pi - \pi]^2. \tag{42}$$

This metric captures the geometric dissimilarity in dihedral angle space, respecting periodicity.

**Time-lagged independent component analysis cost.** The time-lagged independent component analysis (TICA) projection of time-series data captures directions along which the data exhibits maximal autocorrelation. Within molecular dynamics, TICA is commonly used to detect distinct metastable states. Given mean-free time series data $\tilde{x}_t$, the instantaneous (zero-lag) empirical covariance and time-lagged empirical covariance matrix (at lag time $\tau$) are computed as

$$\hat{C}_{00} = \frac{1}{T-\tau} \sum_{t=1}^{T-\tau} \tilde{x}_t \tilde{x}_t^\top, \quad \hat{C}_{0\tau} = \frac{1}{T-\tau} \sum_{t=1}^{T-\tau} \tilde{x}_t \tilde{x}_{t+\tau}^\top. \tag{43}$$

TICA seeks linear projection vectors $w \in \mathbb{R}^n$ that maximize autocorrelation at lag $\tau$

$$\max_w \frac{w^\top \hat{C}_{0\tau} w}{w^\top \hat{C}_{00} w}. \tag{44}$$

The solution to which is obtained by solving the generalized eigenvalue problem

$$C_{0\tau} \lambda = \lambda C_{00} w, \tag{45}$$

where the eigenvalue $\lambda$ measures the autocorrelation of the projected component at lag $\tau$, and the eigenvector $w$ defines the corresponding slow mode. To define the TICA Wasserstein-2 distance TICA-$\mathcal{W}_2$ we take the full evaluation trajectory *without subsampling* and solve Eq. (45) to obtain the first two TICA projection vectors $w_1, w_2$. We may then define the following cost function

$$c_{\text{TICA}}(x,y)^2 = \sum_{j=1}^{2} \left[ w_j^\top x - w_j^\top y \right]^2 \tag{46}$$

defining similarity in TICA projection space. In practice, we compute the TICA projection for the heavy-atom pairwise distances and dihedral angles. We also emphasize that the TICA projection must be computed on the full $5\,\mu s$ evaluation trajectory (such that the slowest transitions may be detected), but that the samples $y$ used in the Wasserstein metric are restricted to the $10^4$ subset.

**Jenson-Shannon divergence metrics.** Following Raja et al. [2025], on the reference MD trajectory we run $k$-means ($k = 20$) on the first two time-lagged independent components (TICs) fitted to the pairwise distances and dihedral angles of the peptide confirmations. Using these clusters, we obtain the occupancy distributions for the reference MD and the samples generated from PROSE, and report their Jenson-Shannon divergence (TICA-$k$-JSD), defined as

$$\text{JSD}(P,Q) = \frac{1}{2}\text{KL}(P,M) + \frac{1}{2}\text{KL}(Q,M), \text{ where } \text{KL}(P,M) = \sum_i P_i \log \frac{P_i}{M_i} \tag{47}$$

where $P$ and $Q$ are discrete distributions, and $M = \frac{1}{2}(P + Q)$. Moreover, we repeat the same procedure using $k$-means only on the dihedral angles, and report this as $\mathcal{T}$-$k$-JSD. In both cases we fit the $k$-means clustering on the features (TICA projections or dihedrals) from full reference MD, but compute the metrics of generated samples against a subsampling of the reference trajectory as previously discussed. Note that JSD depends on the arbitrary choice of clustering (here $k = 20$) and can be sensitive to binning resolution, whereas Wasserstein distances avoid discretization errors by operating directly on the continuous distributions.

### E.3 Sampling algorithm configurations

In this section we define configurations for the sampling algorithms presented in Table 4, as well as the molecular dynamics baseline used in Fig. 1 and Fig. 4. In particular the allocation of the $10^6$ energy evaluations within the method is defined.

**Molecular dynamics baseline.** We follow the same procedure used for collecting the main datasets defined in Appendix B, where the parameters defined in Table 6 are unchanged. We apply a logarithmically decaying frame interval with appropriate reweighting to obtain accurate resolution across many orders of magnitude. The simulation is run for (1 μs) using $10^9$ energy evaluations.

**Sequential Monte Carlo.** For both variants we generate a proposal set of $10^4$ samples. For the discrete variant, we perform 50 annealing steps, requiring two energy evaluations per step: one to update the samples and one for the Metropolis–Hastings correction. For the continuous variants, we perform 100 annealing steps. In both cases, resampling is performed at every step. For the continuous variant, Langevin dynamics is used with a step size $\sigma_t$ of $10^{-7}$ for dipeptides and tetrapeptides, and $\sigma_t$ of $10^{-8}$ for octapeptides. Details of the formulation are found in Appendix D.2. For the discrete variant, we apply Langevin dynamics with a step size of $10^{-5}$, followed by a Metropolis-Hastings step to accept or reject proposals. The step size is adaptively updated to maintain an acceptance rate of approximately 60%, under the assumption of sufficient smoothness in the intermediate densities. Further details of discrete-time AIS are found in Appendix D.3.

**Self-improvement.** We perform 4 rounds of self-improvement. In each round, we spend a portion of the budget to generate $2 \cdot 10^5$ samples and reweight using SNIS. The resulting reweighted samples are then used to finetune the model for 250 gradient steps with a batch size of 256. Notably, once the buffer is established, the finetuning does not further expend the allocated budget as it does not involve energy evaluations. After the final round, the remaining computational budget is allocated to generate a final set of $2 \cdot 10^5$ samples, which are again SNIS reweighted to yield the empirical distribution $\tilde{p}(x|s)$ for a given system $s$. We found it beneficial to introduce an $L_2$ regularization term between the log-densities of the current model and a "teacher" model initialized from the pre-trained weights.

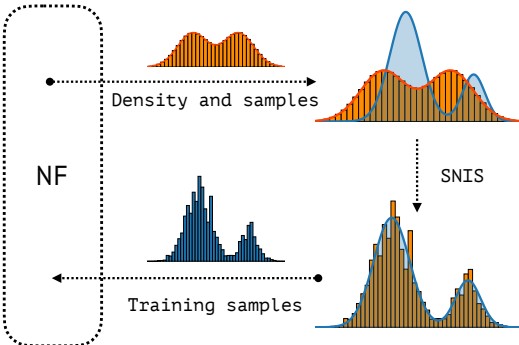

Figure 7: **Self-improvement procedure**. A pre-trained PROSE is finetuned at inference-time by iteratively generating samples, reweighting them using SNIS, and training on the reweighted samples.

### E.4 Additional baseline configurations

**TimeWarp.** The original codebase and model weights of Klein et al. [2023b] were sampled using the asymptotically unbiased MCMC variant (with Metropolis-Hastings acceptance).

**BioEmu.** The inference code of Lewis et al. [2024] was employed. BioEmu does not directly model all-atom resolution hence `hpacker` is employed to introduce the side chains before energy minimization. The codebase of [Lewis et al., 2024] was adapted to use the same Amber14 forcefield as ManyPeptidesMD, on which the $10^4$ energy evaluation budget was spent on per-sample minimization. We additionally experimented with equilibration to target the larger $10^6$ energy budget but could not achieve superior results to minimization-only hence this was not included. The adapted codebase for this baseline is provided at `https://github.com/transferable-samplers/BioEmu`.

**UniSim.** The UniSim model trained on the PepMD dataset of Yu et al. [2025] was evaluated. UniSim applies energy minimization following a proposal step; simulation was ran until the $10^4$ energy budget

was expended. Increasing the simulation time to include the larger $10^6$ energy evaluation budget was not found to improve performance and was omitted. The adapted codebase for this baseline is provided at `https://github.com/transferable-samplers/unisim`.

### E.5 Computational resources

All evaluation experiments are run on a heterogeneous cluster of NVIDIA L40S and RTX8000 GPUs. ECNF++ sampling is parallelized across multiple nodes with unique seeds to reduce sequential runtime. All evaluation timings are recorded using NVIDIA L40S GPUs. The sampling time required for $10^4$ samples for each model is presented in Fig. 8.

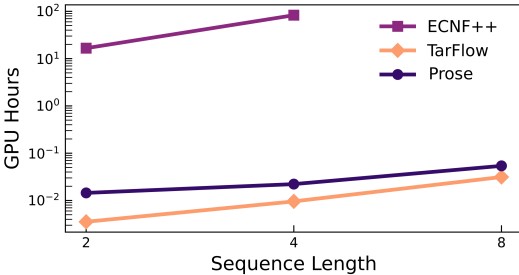

Figure 8: Sampling time for $10^4$ samples on NVIDIA L40S GPU for Boltzmann generators presented in Table 2.

## F Supplementary experimental results

### F.1 Unweighted proposal performance

We compare the performance of ECNF++, TarFlow, and PROSE before and after SNIS (with $10^4$ samples) in Table 11. Evidently, the strongest proposal distribution is given by ECNF++, with both TarFlow and PROSE having large values of $E\text{-}\mathcal{W}_2$ due to high-energy samples. Notably, ECNF++ deteriorates on all metrics after reweighting, suggesting error accumulation in the divergence integration. TarFlow and PROSE improve significantly on $E\text{-}\mathcal{W}_2$ after reweighting; however, only PROSE achieves a reduction in macrostructure metrics through reweighting. Whilst the unweighted ECNF++ proposal achieves stronger $\mathcal{T}\text{-}\mathcal{W}_2$ and TICA-$\mathcal{W}_2$ on dipeptides and tetrapeptides than SNIS-reweighted PROSE, this must be considered with the higher $E\text{-}\mathcal{W}_2$ indicating poor local detail.

Table 11: Quantitative results for flows comparing the proposal performance and performance after importance sampling on peptide systems up to 8 residues. SNIS performed with a budget of $10^4$ energy evaluations.

| Sequence length → | | 2AA (30 systems) | | 4AA (30 systems) | | | 8AA (30 systems) | | |
|---|---|---|---|---|---|---|---|---|---|
| Model ↓ | | $E\text{-}\mathcal{W}_2\downarrow$ | $\mathcal{T}\text{-}\mathcal{W}_2\downarrow$ | $E\text{-}\mathcal{W}_2\downarrow$ | $\mathcal{T}\text{-}\mathcal{W}_2\downarrow$ | TICA-$\mathcal{W}_2\downarrow$ | $E\text{-}\mathcal{W}_2\downarrow$ | $\mathcal{T}\text{-}\mathcal{W}_2\downarrow$ | TICA-$\mathcal{W}_2\downarrow$ |
| ECNF++ | Proposal | 1.958 | 0.143 | 5.006 | 0.582 | 0.335 | — | — | — |
| | SNIS | 3.470 | 0.302 | 10.032 | 1.121 | 0.572 | — | — | — |
| TarFlow | Proposal | $>10^6$ | 0.178 | $>10^9$ | 0.882 | 0.384 | $>10^9$ | 2.475 | 1.026 |
| | SNIS | 0.452 | 0.193 | 1.260 | 0.924 | 0.492 | 11.298 | 2.733 | 1.087 |
| PROSE | Proposal | $>10^9$ | 0.261 | $>10^9$ | 0.916 | 0.546 | $>10^9$ | 2.456 | 1.081 |
| | SNIS | 0.371 | 0.210 | 0.932 | 0.752 | 0.367 | 10.038 | 2.456 | 0.988 |

### F.2 Additional ablations

We consider three further ablations of the PROSE architecture. Firstly, we replace the adaptive conditioning and transition blocks with (i) deeper transformation blocks in which the 8 transformer layers are increased to 20 (ii) a wider transformer block in which the dimension is increased from 384 to 576. In both cases the increased depth / width was defined to approximately match the parameter count of PROSE. We additionally ablate the use of *lookahead conditioning*, in which the atom token $z[i]$ is conditioned not only on $[A_i, R_i, P_i, L_i]$ but also $[A_{i+1}, R_{i+1}, P_{i+1}, L_{i+1}]$, with conditioning information for this pair of indexes mixed using a small MLP. This was motivated by the observation that given naive conditioning the causal masking implies updates to $z[i]$ are computed without

knowledge of $[A_i, R_i, P_i, L_i]$. Preliminary experiments up to tetrapeptides suggested look-ahead conditioning to be beneficial hence it was included in the final PROSE model.

In Table 12 we present results for these ablation models, trained using the same procedure as PROSE. We observe PROSE to marginally outperform both w/o transition variants, while removing the look-ahead conditioning is in fact beneficial, particularly on the shorter sequences. These results invite further research into the optimal allocation of parameters, and advanced conditioning techniques for transferable autoregressive flows.

Table 12: SNIS is performed with a fixed budget of $2 \times 10^5$ energy evaluations.

| Sequence length → | 2AA (30 systems) | | | 4AA (30 systems) | | | | 8AA (30 systems) | | | |
|---|---|---|---|---|---|---|---|---|---|---|---|
| Model ↓ | ESS ↑ | $E$-$\mathcal{W}_2$ ↓ | $\mathcal{T}$-$\mathcal{W}_2$ ↓ | ESS ↑ | $E$-$\mathcal{W}_2$ ↓ | $\mathcal{T}$-$\mathcal{W}_2$ ↓ | TICA-$\mathcal{W}_2$ ↓ | ESS ↑ | $E$-$\mathcal{W}_2$ ↓ | $\mathcal{T}$-$\mathcal{W}_2$ ↓ | TICA-$\mathcal{W}_2$ ↓ |
| PROSE | 0.191 | 0.282 | 0.177 | 0.071 | 0.646 | 0.607 | 0.349 | 0.011 | 9.360 | 2.019 | 0.960 |
| w/o transition (deep) | 0.166 | 0.290 | 0.154 | 0.060 | 0.643 | 0.613 | 0.338 | 0.010 | 9.257 | 2.123 | 0.961 |
| w/o transition (wide) | 0.187 | 0.291 | 0.158 | 0.065 | 0.637 | 0.634 | 0.354 | 0.010 | 9.426 | 2.121 | 0.988 |
| w/o lookahead | 0.212 | 0.270 | 0.158 | 0.074 | 0.591 | 0.623 | 0.372 | 0.011 | 9.319 | 2.070 | 0.962 |

## F.3 Dataset scaling

We explore the effect of dataset size on the performance of PROSE. The full ManyPeptidesMD dataset contains 21,700 sequences simulated for 200 ns each. We train a further four models wherein (i) the trajectories are limited to 50 ns and 100 ns respectively, and (ii) the set of sequences is reduced to 25% and 50%. For the 25% and 50% sequence reduction variants sequence reduction is applied non-linearly with a simple estimate of relative cost, to avoid excessively removing the relatively cheap simulations of short sequences. In Fig. 9 we present metrics for these models. We note the $E$-$\mathcal{W}_2$ to lack any meaningful trend, and the $\mathcal{T}$-$\mathcal{W}_2$ to have limited sensitivity to the dataset variation, despite a slight trend towards larger data subsets improving the metrics. Most interesting of these plots is the TICA-$\mathcal{W}_2$ with a clear trend indicating the full 200 ns trajectories to be beneficial, whilst using only 50% of the sequences is in fact superior to the full dataset. We caveat these results with the observation that PROSE may indeed be operating in the compute-bound regime; even subsampled to 10 ps per frame the ManyPeptidesMD training dataset contains sufficient data for over $8 \cdot 10^5$ training iterations (batch size 512) without repeating a single data sample, in excess of the training budget of $5 \cdot 10^5$ permitted to the PROSE models in this paper.

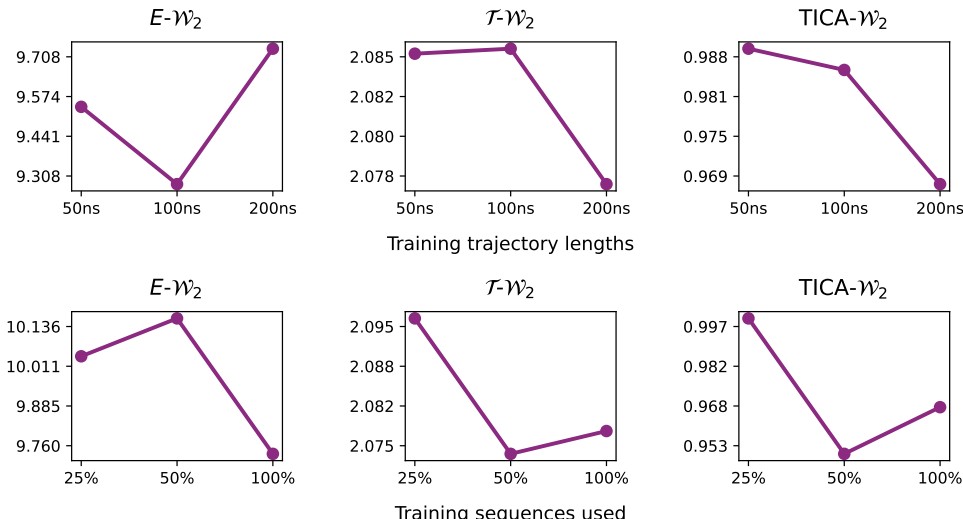

Figure 9: Wasserstein-distance metrics for PROSE trained on variants of ManyPeptidesMD. Upper row: ManyPeptidesMD contains 200 ns trajectories for all training sequences, we train on a variant with 50 ns and 100 ns respectively. Lower: ManyPeptidesMD contains 21,700 sequences, we train using 25% and 50% of the total. Evaluation metrics computed using SNIS on 30 octapeptide sequences.

### F.4 Sequence-length extrapolation

Recall that PROSE is trained exclusively on peptide sequences up to length eight. We explore the capacity of the model to generalize in sequence length beyond its training distribution by evaluating on the nine-residue sequence YQNPDGSQA described by Wu and Brooks [2004] and the well-studied ten-residue small protein Chignolin GYDPETGTWG [Honda et al., 2004]. We additionally evaluate BioEmu and UniSim as baselines on these systems.

Table 13: Comparison of metrics for YQNPDGSQA (9AA) and Chignolin / GYDPETGTWG (10AA).

| | | YQNPDGSQA (9AA) | | | | Chignolin (10AA) | | | |
| --- | --- | --- | --- | --- | --- | --- | --- | --- | --- |
| | | ESS | $E\text{-}\mathcal{W}_2$ | $\mathcal{T}\text{-}\mathcal{W}_2$ | TICA-$\mathcal{W}_2$ | ESS | $E\text{-}\mathcal{W}_2$ | $\mathcal{T}\text{-}\mathcal{W}_2$ | TICA-$\mathcal{W}_2$ |
| UniSim | | – | $>10^5$ | 6.00 | 0.84 | – | 267.68 | 6.48 | 0.20 |
| BioEmu | | – | 160.52 | 4.52 | 1.14 | – | 198.90 | 5.14 | 0.65 |
| PROSE | Proposal | – | $>10^9$ | 3.91 | 1.65 | – | $>10^9$ | 3.63 | 0.96 |
| PROSE | SNIS | 0.0049 | 23.79 | 3.85 | 1.94 | 0.0001 | 832.59 | 4.35 | 1.25 |
| PROSE (self-improve) | Proposal | – | $>10^9$ | 3.73 | 1.95 | – | $>10^9$ | 3.94 | 1.13 |
| PROSE (self-improve) | SNIS | 0.0123 | 18.85 | 3.79 | 1.95 | 0.0002 | 275.87 | 4.43 | 1.20 |

### F.5 JSD evaluation

To further assess distributional alignment between generated samples and samples from the reference MD, we follow Raja et al. [2025] and compute the Jensen–Shannon divergence (JSD) across both TICA projections and backbone torsion angles; for more details on this metric see Appendix E.2

Table 14: Quantitative results for flows comparing the JSD performance on TICA projections and torus angles before and after importance sampling. SNIS performed with a budget of $10^4$ energy evaluations.

| Sequence length → | | 2AA (30 systems) | 4AA (30 systems) | | 8AA (30 systems) | |
| --- | --- | --- | --- | --- | --- | --- |
| Model ↓ | | $\mathcal{T}\text{-}k\text{-JSD}\downarrow$ | $\mathcal{T}\text{-}k\text{-JSD}\downarrow$ | TICA-$k$-JSD $\downarrow$ | $\mathcal{T}\text{-}k\text{-JSD}\downarrow$ | TICA-$k$-JSD $\downarrow$ |
| TimeWarp | — | 0.280 | 0.460 | 0.415 | — | — |
| BioEmu | — | 0.329 | 0.245 | 0.315 | 0.371 | 0.403 |
| UniSim | — | 0.381 | 0.586 | 0.376 | 0.879 | 0.609 |
| ECNF | Proposal | 0.007 | — | — | — | — |
| | SNIS | 0.031 | — | — | — | — |
| ECNF++ | Proposal | 0.002 | 0.004 | 0.004 | — | — |
| | SNIS | 0.020 | 0.051 | 0.052 | — | — |
| TarFlow | Proposal | 0.006 | 0.034 | 0.017 | 0.104 | 0.098 |
| | SNIS | 0.005 | 0.022 | 0.019 | 0.139 | 0.124 |
| PROSE | Proposal | 0.006 | 0.027 | 0.023 | 0.095 | 0.091 |
| | SNIS | 0.004 | 0.011 | 0.009 | 0.109 | 0.082 |

### F.6 Effective sample per second

We report the effective sample size per second (ESS/s) for PROSE and baseline Boltzmann generators, evaluated using SNIS with $10^4$ energy evaluations on an NVIDIA L40s GPU.

Table 15: Effective sample size per second (ESS/s)

| | 2AA | 4AA | 8AA |
| --- | --- | --- | --- |
| ECNF | $1.59 \cdot 10^{-2}$ | — | — |
| ECNF++ | $4.76 \cdot 10^{-3}$ | $2.78 \cdot 10^{-4}$ | — |
| TarFlow | $1.48 \cdot 10^{2}$ | $2.50 \cdot 10^{1}$ | $1.31 \cdot 10^{1}$ |
| PROSE | $7.58 \cdot 10^{1}$ | $1.64 \cdot 10^{1}$ | $9.86 \cdot 10^{-1}$ |

### F.7 Octapeptide Ramachandran plots

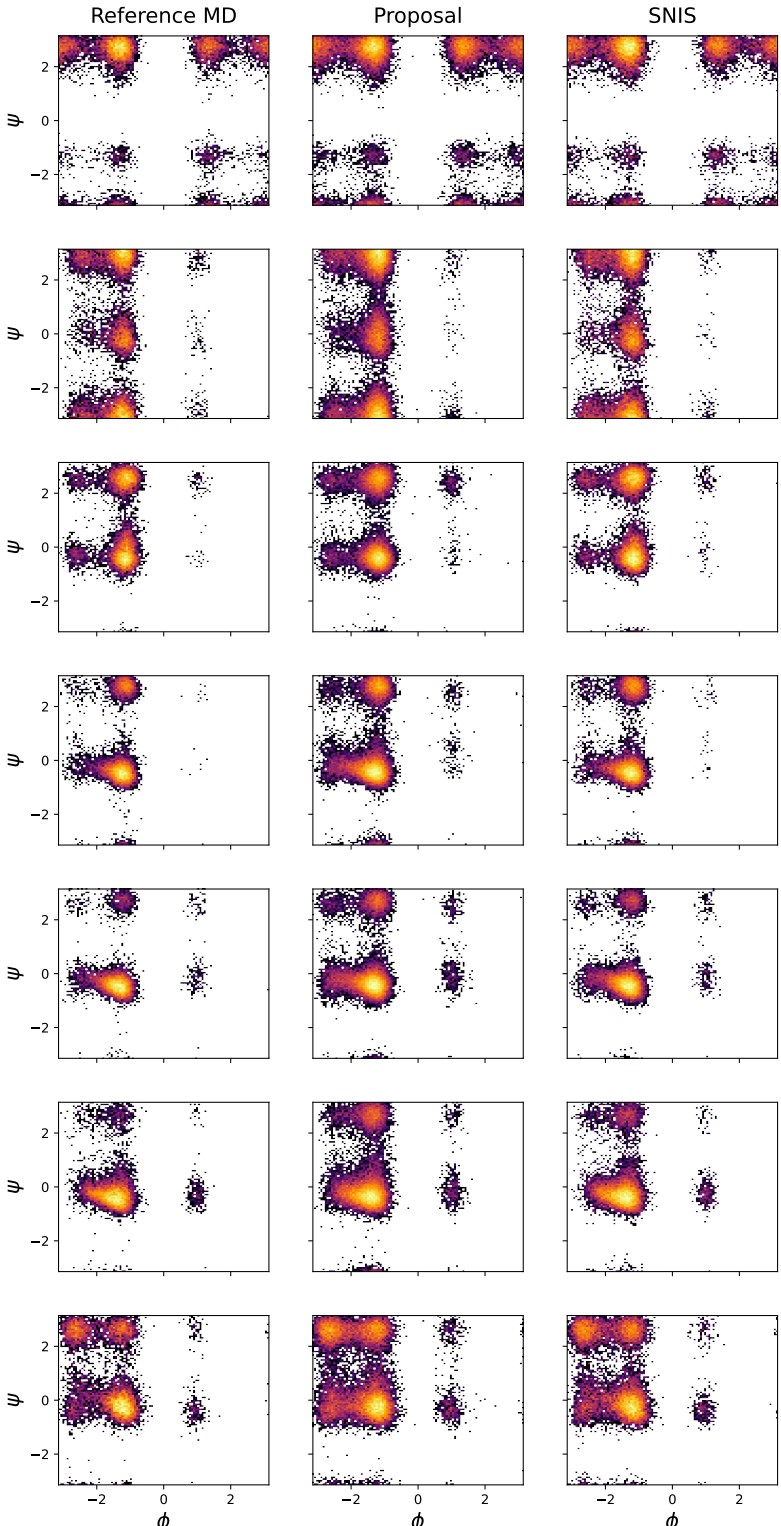

Figure 10: Ramachandran plots for `DGVAHALS` unseen octapeptide system. Reference molecular dynamics (left column), PROSE proposal (center column), PROSE SNIS with $10^5$ samples (right column).

## F.8 Temperature plots

We present TICA plots in Fig. 11 and energy distributions in Fig. 12 across a range of temperatures (310K, 393K, 498K, 631K, 800K). At each temperature, we generate $2 \cdot 10^5$ samples by scaling the prior with the inverse temperature $\beta$, sampling from $\mathcal{N}(0, 1/\beta)$. For SNIS, we use the energy at the corresponding temperature to reweight the samples.

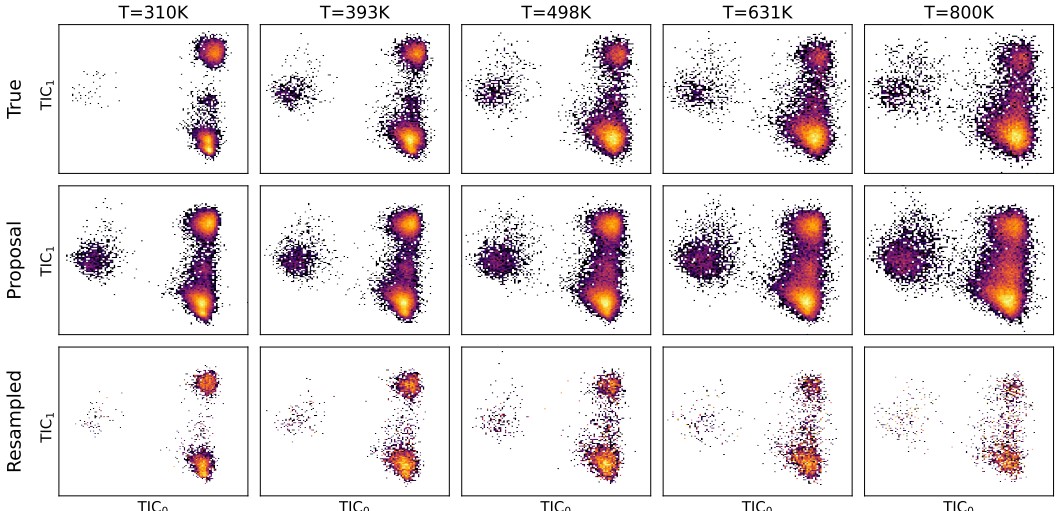

Figure 11: TICA plots for RLMM at different temperatures. Reference molecular dynamics (top row), PROSE proposal (middle row), PROSE SNIS (bottom row) with $5 \cdot 10^4$ samples.

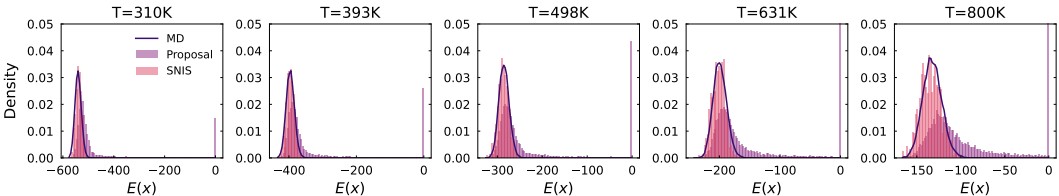

Figure 12: Energy histogram plots for RLMM at different temperatures.

