# OpenReview forum: "Amortized Sampling with Transferable Normalizing Flows"
_NeurIPS.cc/2025/Conference — NeurIPS 2025 poster_

### Official Review · Reviewer_qKai · 2025-06-22

**Clarity:** 2
**Significance:** 3
**Originality:** 3
**Rating:** 4
**Confidence:** 4

**Summary:**

This work introduces ENSEMBLE, an all-atom normalizing flow model designed to be transferable across peptide sequences of varying lengths. The architecture builds upon TarFlow, with key modifications to enable dimensional transferability, adaptive system conditioning, and chemistry-aware sequence permutations. With extensive experiments on a newly curated dataset comprising molecular dynamics trajectories of approximately 16,000 peptide systems, ENSEMBLE consistently outperforms existing deep learning baselines on efficiency and efficacy, and exhibits strong capability for accurate sampling with Monte Carlo algorithms.

**Questions:**

1. In Section 3.1, the description of the model architecture is rather vague. In particular, the concepts of transferability across system dimensions and adaptive system conditioning require clearer and more precise explanations. It would be helpful if the authors could elaborate on these design components in more concrete terms, possibly with supporting equations or schematic illustrations to clarify how they are implemented within the model.
2. The main experimental results presented in Table 2 are not sufficiently comprehensive. It is unclear why the results of ECNF on the 4AA and 8AA test sets, as well as ECNF++ on the 8AA test set, are missing. Even if the model is not inherently adaptable to variable sequence lengths, it is still possible to train it from scratch on datasets with fixed sequence lengths to obtain corresponding evaluation results. This would allow for a fairer and more complete comparison.
3. The authors claim that ENSEMBLE achieves state-of-the-art performance on unseen peptide systems. However, many recent deep learning methods developed for molecular simulation and conformation sampling, such as Timewarp [1], ConfDiff [2], MDGEN [3], AlphaFlow [4], UniSim [5], etc., are not considered. A comparison with these methods is necessary to substantiate the claim.
4. Table 8 shows that ENSEMBLE has significantly more model parameters than the other baselines. This raises the question of whether the observed performance gains are due to increased model size rather than architectural improvements. A more fair comparison, ideally under similar model capacities, would help clarify this point.
5. Table 2 uses ESS as the evaluation metric for sampling efficiency. However, ESS alone does not fully capture inference efficiency, as higher ESS may come at the cost of significantly longer inference time. Following the practice in [1], please report the ESS per second (ESS/s) for each method (including MD trajectories), that is, ESS divided by the wall-clock time required for inference. This would provide a more comprehensive and fair comparison of sampling efficiency across methods.
6. When applying importance sampling using Equation (6), the method requires computing the unnormalized Boltzmann density $p(x)$. Could the authors clarify which force field was used in this computation? Furthermore, please discuss whether the evaluation of $p(x)$ significantly impacts the model’s inference speed. It would be helpful to include relevant statistics to quantify this overhead.
7. It would significantly strengthen the paper to include zero-shot or fine-tuning results of ENSEMBLE on real well-studied protein systems, such as Chignolin. If available, including these results would greatly enhance the practical value and impact of the work.

**Reference**
> [1] Klein, L., Foong, A., Fjelde, T., Mlodozeniec, B., Brockschmidt, M., Nowozin, S., ... & Tomioka, R. (2023). Timewarp: Transferable acceleration of molecular dynamics by learning time-coarsened dynamics. Advances in Neural Information Processing Systems, 36, 52863-52883.

> [2] Wang, L., Shen, Y., Wang, Y., Yuan, H., Wu, Y., & Gu, Q. Protein Conformation Generation via Force-Guided SE (3) Diffusion Models. In Forty-first International Conference on Machine Learning.

> [3] Jing, B., Stark, H., Jaakkola, T., & Berger, B. Generative Modeling of Molecular Dynamics Trajectories. In The Thirty-eighth Annual Conference on Neural Information Processing Systems.

> [4] Jing, B., Berger, B., & Jaakkola, T. AlphaFold Meets Flow Matching for Generating Protein Ensembles. In Forty-first International Conference on Machine Learning.

> [5] Yu, Z., Huang, W., & Liu, Y. (2025). UniSim: A Unified Simulator for Time-Coarsened Dynamics of Biomolecules. In Forty-second International Conference on Machine Learning.

**Ethical Concerns:**

["NO or VERY MINOR ethics concerns only"]

**Final Justification:**

My main concerns were:

- Insufficient baselines: a fair comparison with more recent MD trajectory generation models was needed.

- The generalization ability of ENSEMBLE to larger molecular systems.

- The necessity of using and extending a transformer-based architecture instead of a GNN-based one.

These points were addressed in a fairly detailed manner in the rebuttal. However, the manuscript still lacks clarity in the description of certain implementation details. I suggest the authors revise Section 3.1 to provide a clearer explanation of the model implementation.  Therefore, I will raise my score to 4.

**Limitations:**

As noted in the paper, ENSEMBLE still faces challenges when sampling from lower temperatures. Moreover, the lack of validation on larger and well-studied molecular systems undermines the credibility of the practical applicability of the method.

**Quality:**

3

**Strengths And Weaknesses:**

**Strengths**
- Compared to the backbone model TarFlow, ENSEMBLE introduces improvements in modeling peptide systems, particularly in terms of its ability to generalize across sequences of varying lengths.
- The model supports accurate re-sampling through Monte Carlo algorithms such as Self-Normalized Importance Sampling (SNIS), which also facilitates efficient fine-tuning on previously unseen peptide systems.
- ENSEMBLE demonstrates strong performance in equilibrium sampling tasks on unseen peptide systems, suggesting good generalization capabilities.

**Weaknesses**
- In terms of technical novelty, the contributions appear incremental. The method largely builds on existing architectures (e.g., TarFlow) with relatively minor modifications. The degree of innovation may not be sufficient to justify a major advancement without stronger empirical or theoretical justification.
- The experimental evaluation is insufficient. Some baseline results are incomplete or missing, and the selection of baseline models is limited. A broader and more representative set of comparisons is necessary to convincingly demonstrate the method’s advantages.
- The manuscript lacks clarity in several sections, particularly in the architectural description of ENSEMBLE. Key design choices and their motivations are not always clearly explained, which may hinder reproducibility and understanding by readers from outside the immediate subfield.
- While Transformer-based models like ENSEMBLE must explicitly address issues arising from variable sequence lengths, GNN-based methods are inherently adaptable to sequences of arbitrary length. A more rigorous comparison with representative GNN-based approaches would be important to contextualize the claimed benefits of the proposed architecture.

---

> ### Author Rebuttal · Authors · 2025-07-31
>
> # Response to Reviewer qKai
>
> We would like to thank the reviewer for the time and effort they spent on reviewing our work. We are appreciative of the fact that the reviewer found ENSEMBLE demonstrates "strong performance in equilibrium sampling tasks on unseen peptide systems", and suggests "good generalization capabilities". We now address the concerns raised by the reviewer.
>
> ## Baselines
>
> > However, many recent deep learning methods developed for molecular simulation and conformation sampling, such as Timewarp [1], ConfDiff [2], MDGEN [3], AlphaFlow [4], UniSim [5], etc., are not considered. A comparison with these methods is necessary to substantiate the claim.
>
> We thank the reviewer for suggesting these additional baselines on conformation sampling. We include additional baselines in the table below, using the same 1e4 energy evaluation budget as in Table 1 of the main paper.
>
> |           | (2AA)          |             | (4AA)          |             |             | (8AA)         |             |             |
> |-----------|----------------|-------------|----------------|-------------|-------------|---------------|-------------|-------------|
> |           | E-W2           | Torus-W2    | E-W2           | Torus-W2    | TICA-W2     | E-W2          | Torus-W2    | TICA-W2     |
> | Ensemble  | 0.523          | 0.379       | 1.565          | 2.051       | 0.522       | 12.201        | 5.354       | 1.047       |
> | TimeWarp  | 5.765          | 1.515       | 6.436          | 3.448       | 1.203       |       --        |     --        |        --     |
> | UniSim    | 176882.028     | 1.052       | 30498.199      | 2.860        | 2.112       | 36559.357     | 4.813       | 1.704       |
> | BioEmu    | 2622.421       | 0.959       | 519.781        | 2.538       | 1.168       | 159.524       | 4.98        | 1.14        |
>
> In each case we use the pretrained model weights and sampling configurations provided by the original authors. In the case of Timewarp the model only supports peptides of lengths 2AA and 4AA hence no results are presented for 8AA.  Following the authors procedure, for BioEmu the sidechains are added using hpacker followed by energy minimization. Additionally, for UniSim we employ the energy minimization step during the trajectory generation. In both cases a sweep was performed to optimize the energy minimzation budget allocated per-sample with the number of samples generated.
>
> We would like to emphasize that only Ensemble (with SNIS), ECNF (with SNIS), and Timewarp (with Metropolis-Hastings acceptance) are able to draw asymptotically unbiased samples from the target distribution. Empirically we observe an a significant improvement of ENSEMBLE over baselines with respect to all metrics, except the macrostructure (Torus-W2 and TICA-W2) metrics on the largest systems (8AA). We however, emphasize that the E-W2 achieved by Ensemble to be significantly lower on the 8AA systems than the next best method.
>
> ## Practical applicability
>
> > Moreover, the lack of validation on larger and well-studied molecular systems undermines the credibility of the practical applicability of the method.
>
> We would like clarify that these are by far the largest systems that have been studied in a transferable setting for Boltzmann generator type models - a 4-fold increase given the only prior transferable Boltzmann generator operated on dipeptides only. We believe our work therefore significantly enhances the practical value and impact of this model class. While Boltzmann Generators as a model class are inherently less scalable than vanilla generative models, they offer the key advantage of exact likelihoods for rigorous statistical re-weighting. We argue that solving the fundamental challenge of length-transferability for peptides up to 8 residues is a critical and necessary step forward, establishing the foundation needed before this rigorous approach can be scaled to larger, canonical protein systems. Our work pushes the practical frontier for this entire class of models, making the future application to larger systems a more tangible research goal.
>
> ## Evaluation
>
> > please report the ESS per second (ESS/s) for each method (including MD trajectories)
>
> In the following table we report effective samples per second for ENSEMBLE and baseline methods. We will report ESS/s for MD trajectories and the additional baseline methods during the discussion period.
>
> |          | 2AA       | 4AA       | 8AA       |
> |--|--|--|--|
> | ECNF     | 3.20E-02  | --         | --         |
> | ECNF++   | 3.89E-03  | 3.17E-04  | --         |
> | TarFlow  | 8.44E+01  | 9.44E+00  | 5.56E-01  |
> | Ensemble | 2.20E+01  | 8.06E+00  | 5.00E-01  |
>
> > This raises the question of whether the observed performance gains are due to increased model size rather than architectural improvements.
>
> We thank the reviewer for their concern. We would like to clarify that in our view the model size increase is actually a substantial benefit of ENSEMBLE over previous Boltzmann generator type approaches. Specifically, because our likelihood computation is efficient, we can scale the model significantly more without computational difficulties that might severely limit sampling efficiency. We further draw attention to the sampling wall-time results presented in Figure 22 of the supplementary material, which indicate ENSEMBLE to be significantly faster to sample from than the ECNF, with diminishing increase in wall-time over the standard TarFlow as the system size is increased.
>
> ## Chignolin
>
> Following the reviewer's suggestion, in the table below, we report the performance of the method on the 9-residue peptide YQNPDGSQA introduced by [1] and the well-studied Chignolin small protein. The reported metrics demonstrate that the proposed method outperforms all of the baselines. We further explore an additional step in the self-refinement procedure where the first round of proposal samples are not reweighed using SNIS, but instead undergo constrained energy minimization before they are used for self-refinement training. The self-refined model is then used to generate proposal samples which are reweighed using SNIS. Note that the training data only contains systems of length up to 8 amino acids, making clear the ability of Ensemble to extrapolate beyond it's training data distribution, despite the baselines being trained on larger systems.
>
> |  | YQNPDGSQA (9AA) |  |  |  | Chignolin (10AA) |  |  |  |
> | :- | - | :- | :- | :- | - | :- | :- | :- |
> |  | ESS | E-W2 | Torus-W2 | TICA-W2 | ESS | E-W2 | Torus-W2 | TICA-W2 |
> | Ensemble (SNIS) | 0.0062 | 24.78 | 3.73 | 1.39 | 0.0001 | 829.59 | 4.21 | 0.73 |
> | Ensemble (self-refine + SNIS) | 0.0082 | 21.34 | 3.77 | 1.39 | 0.0003 | 447.72 | 4.64 | 1.002 |
> | Ensemble + constrained energy min + self-refine | \- | \- | \- | \- | 0.0002 | 407.13 | 4.52 | 0.81 |
> | BioEmu | - | 1629.55 | 3.85 | 1.03 | - | 9.74e9 | 3.74 | 0.89 |
> | UniSim | - | 1630.42 | 5.65 | 1.05 | - | 3.21e9 | 6.06 | 1.26 |
>
> ## Novelty
>
> We thank the reviewer for this crucial feedback. We argue our technical novelty lies not in creating a flow from scratch, but in solving a long-standing challenge: **enabling a single, all-atom model with efficient likelihood calculation to transfer zero-shot across variable-length peptides**
>
> This required novel architectural solutions for **dimensional transferability and adaptive conditioning**, which are non-trivial extensions that allow a Transformer-based flow to generalize across system sizes for the first time. The significance of this breakthrough is validated by our empirical results, where ENSEMBLE unlocks a previously unattainable capability and achieves a >4000x speedup over transferable baselines. We will revise the paper to frame our contribution more clearly around solving this specific, challenging problem.
>
> ## Clarity
>
> > Could the authors clarify which forcefield was used in this computation?
>
> The amber-14 forcefield was used through the OpenMM package. For details on the simulation please see Table 5 in the appendix.
>
> Yes, energy computation has a significant impact on the model's inference speed, this is why we set a fixed number of energy evaluations budget across all methods. While we use a relatively efficient energy function here, we note that our method can easily be adapted to any energy function, some of which may be significantly more expensive than even our fairly large ENSEMBLE model to run. Given the favourable performance of ENSEMBLE with respect to energy evaluations demonstrated in Figure 1, any increase in energy function complexity can be anticipated to translate into further-still wall-time performance advantages.
>
> ## Concluding remarks
>
> We thank the reviewer again for their time. We believe we have answered, to the best of our ability, all the great questions raised by the reviewer. We hope our answers allow the reviewer to consider potentially upgrading their score if they see fit. We are also more than happy to answer any further questions.
>
> [1] Xiongwu Wu and Bernard R Brooks. "Beta-hairpin folding mechanism of a nine-residue peptide revealed from molecular dynamics simulations in explicit water." _Biophys J._ (2004)

---

> > ### Comment · Reviewer_qKai · 2025-08-01
> >
> > Thank you to the authors for the detailed response, which addressed some of my concerns. Below are the issues I still retain:
> >
> > 1. I greatly appreciate the inclusion of additional baselines such as Timewarp, UniSim, and BioEmu. Introducing state-of-the-art deep learning models tailored for MD trajectory generations indeed strengthens the work's credibility, and the performance of ENSEMBLE is also impressive. That said, it is important to point out that all three baselines are time-dependent trajectory generation models, in contrast to independently-sampled Boltzmann generators. This fundamental difference in sample generation may explain why those baselines tend to produce samples with significantly higher energy. To ensure a fairer comparison, I recommend that the authors also include comparisons to other Boltzmann generators, such as TBG [1].
> >
> > 2. On novelty: I agree that the proposed method addresses a real challenge, namely, that transformer-based flows struggle to generalize across molecular systems of varying sizes. However, my view is that this issue is naturally resolved by GNN-based approaches, which are invariant to atom order and flexible in system size. Therefore, the necessity of using and extending a transformer-based model in this context requires further justification.
> >
> > 3. The authors did not respond to my earlier concern regarding the clarity of the exposition in Section 3.1. For example, the sentence “We therefore define appropriate masking to the affine sequence updates and log-determinant aggregation to prevent padding tokens influencing either computation, under arbitrary sequence permutations” lacks specificity. I would appreciate a clearer and more detailed explanation of how this masking is actually designed.
> >
> > 4. Regarding the model size, I acknowledge the authors' point that ENSEMBLE achieves orders-of-magnitude faster inference compared to CNF, even with significantly more parameters. However, this does not change the fact that the comparison between ECNF and ENSEMBLE was made under vastly different model sizes. In particular, Table 2 shows that ECNF achieves better performance on the 2AA peptide system with fewer parameters, which leaves open the question of whether the architectural advantage of ENSEMBLE is truly significant.
> >
> > If the authors could further address these points, I would be willing to raise my score.
> >
> > > [1] Klein, Leon, and Frank Noé. "Transferable Boltzmann Generators." Advances in Neural Information Processing Systems 37 (2024): 45281–45314.

---

> > > ### Author Response · Authors · 2025-08-03
> > >
> > > We sincerely thank the reviewer for the prompt response and continued engagement. Below, we address the remaining concerns:
> > >
> > > ## Comparison to TBG
> > >
> > > We are grateful for the reviewers comment that providing the additional baselines "strengthens the work's credibility". Regarding the request for another uncorrelated sampler e.g TBG [1], we would like to clarify that the equivariant continuous normalizing flow (ECNF) baseline provided in the paper is exactly the TBG model weights as trained by [1]. While mentioned in section 4.2, we will revise the manuscript to clarify this. Furthermore, we would like to emphasize that the vector field in ECNF is parameterized by an equivariant GNN, hence making this a GNN-based method.
> > >
> > > ## Justification of Transformer
> > >
> > > We acknowledge the reviewer's comment that GNN-based models are adaptable to varying sizes of molecular systems. We firstly remark that transformers themselves can be formulated as attention-based GNNs operating on fully connected graphs, and without positional encodings are permutation equivariant, and naturally extend to sequences of arbitrary length. While the TarFlow was designed for systems of fixed-length [2], the transformer architecture itself is a well-motivated backbone for an autoregressive flow, as molecular systems of varying size simply correspond to sequences of varying length.
> > >
> > > Our framework builds upon the TarFlow [2, 3], where strong performance has been demonstrated in both the image and molecular domains. A central goal of our framework is to combine efficient likelihood computation with architectural scalability. In contrast, GNNs, with their unstructured sparsity patterns, are often more difficult to scale and do not share the same hardware optimizations enjoyed by transformers, making them less suitable for large, likelihood-based models. To the best of our knowledge no normalizing flow (GNN-based or otherwise) prior to [2] has demonstrated comparable scalability, hence motivating our choice of TarFlow as a starting point for our framework.
> > >
> > > ## Masking Details
> > >
> > > We thank the reviewer for raising this point. During training, for a given maximum system size M, each system with $N_s$ atoms is padded from shape ($N_s$, 3) to (M, 3) using ($M - N_s$, 3) zero-valued tokens to ensure all systems have the same size for efficient training. Alongside this, we maintain a binary mask that distinguishes the real atom tokens ($mask_i=1$) from the padding tokens ($mask_i=0$). This mask is used throughout the transformer-based normalizing flow to ensure that:
> > >
> > > 1. Affine transformations do not modify the padding tokens (e.g equations (3) and (4) of [2] are no-ops where $mask_i = 0$)
> > >
> > > 2. When aggregating log-determinant (e.g equation (6) of [2]) we exclude contributions from the padding tokens where $mask_i=0$
> > >
> > > Naively, when using the "flip" variants of permutations the padding tokens would be placed at the start of the sequence. However, our implementation ensures that the padding tokens are always appended to the ends of sequences for all permutations. Given the positional encodings applied to the sequence, this is necessary to achieve the exact same outputs irrespective of the use of (or length of) padding. Crucially, this enables exact invertibility between the reverse pass in training (where sequences are padded), and the forward pass during proposal generation which does not support leading padding due to the KV-cache based implementation.
> > >
> > > We thank the reviewer for highlighting the need to better communicate this and will update the manuscript accordingly.
> > >
> > > ## Performance on 2AA and Model Size
> > >
> > > We acknowledge the reviewer's concern regarding the performance difference between ECNF and Ensemble on the 2AA task. Indeed, ECNF performs well on this small system despite its smaller model size. However, we assert that the notable difference in ESS must be considered with respect to the sampling walltime. In the ESS/s results requested by the reviewer, Ensemble achieves 3 orders of magnitude higher ESS/s than ECNF. The lower E-W2 and Torus-W2 for ECNF follows trends observed by [3], where ECNF variants performed well on alanine dipeptide, but degrade with larger systems.
> > >
> > > We further clarify that ECNF is trained specifically on 2AA, whereas Ensemble is trained on a broader set of peptides up to 8AA. This added generalization complexity is absent in ECNF. Despite this, our method still performs comparably.
> > >
> > > ## Concluding Remarks
> > >
> > > We thank the reviewer again for their proactive engagement, and their willingness to raise their score. We hope our response has addressed their remaining concerns and remain available for further clarification.
> > >
> > > [1] Klein, Leon, and Frank Noé. "Transferable Boltzmann Generators." Advances in Neural Information Processing Systems 37 (2024): 45281--45314.
> > >
> > > [2] Zhai et al. "Normalizing Flows are Capable Generative Models" ICML (2025)
> > >
> > > [3] Tan et al. "Scalable Equilibrium Sampling with Sequential Boltzmann Generators" ICML (2025)

---

> ### Comment · Reviewer_qKai · 2025-08-03
>
> Thank you for the prompt response. Most of my concerns have been addressed, and I have a few minor suggestions remaining:
>
> - I agree with the author’s explanation that the transformer architecture offers better scalability compared to GNN-based models. I hope the author will later support this point with actual references and evidence.
>
> - The author mentions in the rebuttal that the performance of the ECNF model based on the GNN architecture will degrade with larger systems. However, from Table 2, it can be observed that the performance of ENSEMBLE also deteriorates as the system size increases. Additionally, the experimental results for ECNF on 4AA and 8AA are missing in Table 2, which makes it difficult to compare the performance degradation of ECNF and ENSEMBLE on larger molecular systems. This undermines the necessary argumentation for the author’s viewpoint. I believe this point requires further clarification.

---

> > ### Author Response · Authors · 2025-08-04
> >
> > We thank the reviewer for their prompt response, and are pleased the reviewer feels “most of my concerns have been addressed” and that there are only minor suggestions remaining.
> > > I agree with the author’s explanation that the transformer architecture offers better scalability compared to GNN-based models. I hope the author will later support this point with actual references and evidence.
> >
> > We additionally thank the reviewer for suggesting references to support our use of transformer architectures as a scalable alternative to GNN-based methods [1]. In particular, transformers have demonstrated superior scalability and performance in biological applications [2,3], partly due to their alignment with modern hardware. We will clarify this point and strengthen the supporting citations in the revised manuscript.
> >
> > > The author mentions in the rebuttal that the performance of the ECNF model based on the GNN architecture will degrade with larger systems. However, from Table 2, it can be observed that the performance of ENSEMBLE also deteriorates as the system size increases.
> >
> > We would like to clarify we meant that ECNF based models degrade in performance with larger systems **relative to transformer-based normalizing flows** as seen in Table 9 and Table 10 of [4]. We agree that all models including ENSEMBLE have a harder time as the system size increases, which makes sense as larger systems are significantly more difficult on average.
> >
> > > Additionally, the experimental results for ECNF on 4AA and 8AA are missing in Table 2, which makes it difficult to compare the performance degradation of ECNF and ENSEMBLE on larger molecular systems. This undermines the necessary argumentation for the author’s viewpoint. I believe this point requires further clarification.
> >
> > We would like to clarify that we argue that **ECNF is computationally infeasible at 8AA scale** and do not argue the relative performance between ECNF and ENSEMBLE at that scale. To support this argument we note that **to evaluate ECNF on 8AA would cost more than** \$20000 **vs.** \$ 7 **for ENSEMBLE** in compute. For further detail we provide an approximate cost table below for sampling at the 4AA and 8AA scale. This table is calculated using sampling wall times on an NVIDIA L40S GPU and using current pricing on amazon web services. 8AA time is extrapolated from
> > | | 4AA | 8AA |
> > |-|-|-|
> > | ECNF | $2000 | >$20000 |
> > | ENSEMBLE | $4 | $7 |
> >
> > We would also like to clarify that **a strictly superior model to ECNF is already presented in Table 2 on 4AA**. As shown in Table 3 of [4]. ECNF++ outperforms ECNF in all settings. The reason that ECNF outperforms ECNF++ in our Table 2 is that it is trained only on a single (2AA) system size following TBG [2]. Therefore the conclusion from this experiment is that single system size training is easier to generalize over than multi system size training. The ECNF number is taken from [5], whereas we train the improved model ECNF++ as a stronger baseline.
> >
> > We hope this clarifies our argument. We once again thank the reviewer for their continued engagement, and hope our response has addressed their remaining minor concerns.
> >
> > ### References
> > [1] Chaitanya K. Joshi. “Transformers are Graph Neural Networks” arxiv.2506.22084 (2025)
> >
> > [2] Abramson, J., Adler, J., Dunger, J. et al. Accurate structure prediction of biomolecular interactions with AlphaFold 3. Nature 630, 493–500 (2024).
> >
> > [3] Geffner, Tomas, et al. "Proteina: Scaling flow-based protein structure generative models." arXiv preprint arXiv:2503.00710 (2025).
> >
> > [4] Tan et al. “Scalable Equilibrium Sampling with Sequential Boltzmann Generators” ICML (2025)
> >
> > [5] Klein, Leon, and Frank Noé. "Transferable Boltzmann Generators." Advances in Neural Information Processing Systems 37 (2024): 45281–45314.

---

> > > ### Comment · Reviewer_qKai · 2025-08-05
> > >
> > > Thanks for the response. My concerns have been addressed, and I will raise my score.

---

> > > > ### Author Response · Authors · 2025-08-05
> > > >
> > > > We are grateful for the reviewer’s active engagement and suggestions which greatly improved our work. We are glad they feel their concerns have been addressed, and are appreciative of their intention to raise this score to reflect this. We once again thank the reviewer for their continued, diligent attention to our work.

---

### Official Review · Reviewer_7hoH · 2025-06-27

**Clarity:** 3
**Significance:** 3
**Originality:** 2
**Rating:** 4
**Confidence:** 4

**Summary:**

The paper demonstrates that transformer-based normalizing flows, trained on MD simulations of thousands of peptides, can generalize to new peptides and serve as exact-likelihood importance samplers of their Boltzmann distributions.

**Questions:**

See above.

**Ethical Concerns:**

["NO or VERY MINOR ethics concerns only"]

**Final Justification:**

Authors provided new experimental details to transparently convey the performance of the method.

**Limitations:**

Yes

**Paper Formatting Concerns:**

Looks good!

**Quality:**

3

**Strengths And Weaknesses:**

**Strengths**
The paper is the first demonstration of a Boltzmann generator that is transferable, evaluated on unseen peptides, and has fast and tractable likelihoods so as to permit efficient importance sampling. This will stand as a important milestone in the field.

**Weaknesses**
While the result is conceptually significant, the paper leaves a lot of questions unanswered and it is hard to judge the performance of the method in a rigorous context. It is the omission of these details in itself that detracts from the paper, not the results - the authors should optimize for transparency rather than hiding results that qualify the significance.
* Figure 1. If test peptides have been simulated for up to 1us, then the MD results should be available for wall-clock times up to O(10) GPU-hrs, but authors have concealed this part of the curve.
* It is very hard to conceptualize TICA and T-W2s. Authors should use JSDs or build markov state models and report recovery of markov states.
* With very small ESS, it is unclear if SNIS is effective at estimating observables better than the unweighted proposal with small sample sizes. Authors should also report results without reweighting.
* It seems that the model still generates many high-energy samples. Authors should report statistics - what are the mean, median, 95th percentile energies of the unweighted samples? How does this change from tetrapeptides to octopeptides? This is important to understand the further scalability of the method.
* It would be helpful to understand the impact of scaling the training data from 1k to 10k training peptides. There is a missed opportunity to explore scaling laws wrt both longer simulations and more training peptides.
* Authors should show results on chignolin - the only peptide of nontrivial length considered to have been simulated to convergence - since it is a common reference point with many other works.

---

> ### Author Rebuttal · Authors · 2025-07-31
>
> # Response to Reviewer 7hoH
>
> We thank the reviewer for their time and effort in reviewing our work. We are pleased that the reviewer describes this paper as "the first demonstration of a transferable Boltzmann generator" and as "an important milestone in the field." In what follows, we address the concerns raised and provide suggested evaluations.
>
> > Figure 1. If test peptides have been simulated for up to 1us, then the MD results should be available for wall-clock times up to O(10) GPU-hrs, but authors have concealed this part of the curve.
>
> Figure 1 does not contain MD trajectories up to 1$\mu$s; instead, the baseline MD trajectories there are simulated up to a fixed energy evaluation budget ($10^6$ energy evaluations). Note that simulating 1$\mu$s would require $10^9$ energy evaluations. The long MD simulations (1$\mu$s long) are used only as ground truth samples for metrics evaluation, and use a less frequent frame storing interval to prevent excessive data accumulation (making them unsuitable for use as a baseline across many orders of magnitude of energy evaluation budget). To avoid potential confusion, in the text, we will refer to different types of MD trajectories as train MD, eval MD, and ground true MD, which are used, respectively, as training data, as a baseline, and as the ground true distribution of samples.
>
> However, we agree with the reviewer that the "stopping" of the MD baseline in Figure 1 is a missed opportunity for comparison between our method and MD at larger wall-time budgets. Hence we have rerun the MD baseline using a decaying frame storing interval up to 1e8 energy evaluations. Given we cannot communicate an updated figure, in the below table we present the metrics at various wall time budgets, in comparison to ENSEMBLE, where a spline was fit to enable arbitrary wall times to be queried.
>
> | L40 Hours |      E-W2      |           |     Torus-W2    |           |     TICA-W2    |           |
> |--|--|--|--|-- |-- |-- |
> |           | MD Baseline    | Ensemble  | MD Baseline     | Ensemble  | MD Baseline    | Ensemble  |
> | 4.41e-4 | 6.764          | 4.400       | 2.734           | 3.252     | 1.192          | 1.025     |
> | 3.09e-3  | 2.490           | 2.585     | 2.044           | 2.663     | 1.238          | 0.769     |
> | 2.95e-2 | 1.445          | 1.356     | 1.605           | 2.057     | 1.166          | 0.571     |
> | 2.94e-1  | 0.861          | 0.964     | 1.556           | 1.672     | 0.983          | 0.494     |
> | 2.93e0 | 0.611          | 1.323     | 1.56            | 1.673     | 0.825          | 0.558     |
>
> > It is very hard to conceptualize TICA and T-W2s. Authors should use JSDs or build markov state models and report recovery of markov states.
>
> Following the reviewer's suggestion, we're currently running the evaluation using JSD and will share the updated results during the discussion period.
>
> > With very small ESS, it is unclear if SNIS is effective at estimating observables better than the unweighted proposal with small sample sizes. Authors should also report results without reweighting.
>
> The suggested comparison is provided in Table 10 (Appendix C.1). This comparison demonstrates that the proposal samples multiple outliers (unphysical structures having very high values of energy), which are efficiently filtered out via SNIS.
>
> > It seems that the model still generates many high-energy samples. Authors should report statistics - what are the mean, median, 95th percentile energies of the unweighted samples? How does this change from tetrapeptides to octopeptides? This is important to understand the further scalability of the method.
>
> We thank the reviewer for the excellent point. We provide the requested statistics for the unweighted proposal samples for sequence lengths of 4 and 8.
>
> | seq_length | mean          | median        | percentile_95  |
> |------------|---------------|---------------|----------------|
> | 4          | 8.337811e+11  | 2.436593e+05  | 2.531143e+11   |
> | 8          | 1.154441e+18  | 9.880774e+13  | 2.782726e+18   |
>
> The results highlight a notable increase in energy with longer sequences, reflecting the greater complexity of the larger system. We believe this further motivates the need for reweighting strategies at higher sequence lengths.
>
> > It would be helpful to understand the impact of scaling the training data from 1k to 10k training peptides. There is a missed opportunity to explore scaling laws wrt both longer simulations and more training peptides.
>
> We thank the reviewer for their suggestion. We agree that this could be quite useful for future work. Given the computational requirement of proper scaling plots in this domain, we will add this experiment in the camera-ready version of the paper.
>
> > Authors should show results on chignolin - the only peptide of nontrivial length considered to have been simulated to convergence - since it is a common reference point with many other works.
>
> Following the reviewer's suggestion, in the table below, we report the performance of the method on the 9-residue peptide YQNPDGSQA introduced by [1] and Chignolin. The reported metrics demonstrate that the proposed method outperforms all of the baselines. Note that the training data only contains systems of length up to 8 amino acids, which suggests that the method generalizes across systems of previously unseen length.
>
> |  | YQNPDGSQA (9AA) |  |  |  | Chignolin (10AA) |  |  |  |
> | :- | - | :- | :- | :- | - | :- | :- | :- |
> |  | ESS | E-W2 | Torus-W2 | TICA-W2 | ESS | E-W2 | Torus-W2 | TICA-W2 |
> | Ensemble (SNIS) | 0.0062 | 24.78 | 3.73 | 1.39 | 0.0001 | 829.59 | 4.21 | 0.73 |
> | Ensemble (self-refine + SNIS) | 0.0082 | 21.34 | 3.77 | 1.39 | 0.0003 | 447.72 | 4.64 | 1.002 |
> | Ensemble + constrained energy min + self-refine | \- | \- | \- | \- | 0.0002 | 407.13 | 4.52 | 0.81 |
> | BioEmu | - | 1629.55 | 3.85 | 1.03 | - | 9.74e9 | 3.74 | 0.89 |
> | UniSim | - | 1630.42 | 5.65 | 1.05 | - | 3.21e9 | 6.06 | 1.26 |
>
> We thank the reviewer again for the detailed feedback and constructive suggestions on the improvement of the manuscript. We hope we address their main concerns, and we remain available throughout the discussion period for further clarifications and suggestions for improvement.

---

> > ### Comment · Reviewer_7hoH · 2025-08-06
> >
> > I appreciate the authors' comprehensive response! I look forward to seeing the JSD and MSM results in the discussion period, as authors' have indicated. Upon receiving those results, and if authors agree to include all rebuttal results in the revision, I would be happy to raise the score. Thank you!

---

> > > ### Comment · Area_Chair_uubj · 2025-08-08
> > >
> > > @Reviewer 7hoH: The authors have presented the results you requested. Please respond whether this answers your questions, and if you'd like to update your score.

---

> ### Author Response · Authors · 2025-08-07
>
> We sincerely thank the reviewer for their thoughtful feedback and for their willingness to raise their score based on the additional results. We appreciate the constructive suggestions, which have helped strengthen the quality of our work.
>
> > Upon receiving those results, and if authors agree to include all rebuttal results in the revision
>
> We confirm that we will include all rebuttal results in the revised version of the manuscript. We thank the reviewer for emphasizing this point and for encouraging a more complete presentation of our findings.
>
> >  I look forward to seeing the JSD and MSM results in the discussion period
>
> As requested, we report below the Jensen-Shannon Divergence (JSD) results for the 2AA, 4AA, and 8AA systems. These results assess the recovery of discrete states by our generative model compared to reference MD simulations. We follow the procedure outlined in [1]. Specifically:
>
> - We perform k-means clustering on the reference MD trajectories using the top 2 time-independent components (TICs), which are computed from a featurization of pairwise distances and dihedral angles.
> - The MD trajectories are clustered into 20 discrete states.
> - Using the fitted k-means model, we assign cluster labels to both the reference MD and the generated samples.
> - We then construct histograms over these cluster assignments and compute the JSD between the reference and generated distributions.
> - We report this JSD metric for all methods, including new baselines requested by reviewer qKai in the following table, where all methods are given a budget of 10^4 energy evaluations.
>
> | | 2AA | 4AA | 8AA |
> |--|--|--|--|
> | ECNF | **0.0022** | -- | -- |
> | ECNF++ | 0.0192 | 0.0549  |    --     |
> |  TarFlow  | 0.0080  | 0.0299  | 0.1727  |
> |  Ensemble  | 0.0062  | **0.0253**  | **0.1279**  |
> | MD Baseline | 0.3660  | 0.3638  | 0.5259  |
> |  BioEmu    | 0.3061  | 0.1769  | 0.2534  |
> | TimeWarp | 0.2107  | 0.3768  |     --    |
> | UniSim           | 0.3774  | 0.4005  | 0.5371  |
>
> This metric aligns with the results of Table 2 in the main paper, in which ECNF (trained only on 2AA) is the strongest method on the simplest (2AA) test dataset, but Ensemble is the strongest method on the more challenging 4AA and 8AA test datasets. The outperforming of ECNF on the 2AA dataset can be attributed to the simpler task (without sequence length transferability), and we note the ECNF++ (trained on 2AA -> 4AA) to be notably inferior on both 2AA and 4AA compared to Ensemble. These results will also be included in the updated manuscript.
>
> Concerning the reviewers request of MSM results, while exploring this metric we found that it is not possible to compute an MSM metric for Ensemble, Tarflow, ECNF, and Boltzmann Generator methods in general. The work of [1] reports two additional metrics based on the MSM transition matrix fit on these discrete states: “Transition Negative Log Likelihood” and “Fraction of Valid Paths”. We clarify that Ensemble (as well as TarFlow and ECNF), generate uncorrelated proposal samples which are then reweighted using e.g. SNIS. This implies there is no notion of “transitions” between states, and therefore it is not possible to compare to a reference transition matrix as in [1]. This is in contrast to methods such as MD, TimeWarp, UniSim, which generate samples in a time-sequential manner and hence could be evaluated in such a way.
>
> We once again thank the reviewer for their time reviewing our work, and their insightful suggestions for improvement.
>
> [1] Raja, Sanjeev, et al. "Action-Minimization Meets Generative Modeling: Efficient Transition Path Sampling with the Onsager-Machlup Functional." arXiv preprint arXiv:2504.18506 (2025).

---

> > ### Comment · Reviewer_7hoH · 2025-08-08
> >
> > Thank you for these results. Actually, the discrete state JSD metrics you have reported corresponds to what I meant by MSM metrics. Could you provide JSD metrics for the continuous collective varies, i.e., torsion angles? I will assume in good faith those results will be provided and will raise the score.

---

> > > ### Author Response · Authors · 2025-08-09
> > >
> > > We are glad to have addressed the reviewers request with the provided JSD metric. We will include the additionally requested torsion angle JSD metric in updated versions of the work. We sincerely thank the reviewer for their time spent reviewing our work, and for raising their score of their review.

---

### Official Review · Reviewer_EUPF · 2025-07-03

**Clarity:** 3
**Significance:** 3
**Originality:** 2
**Rating:** 4
**Confidence:** 3

**Summary:**

This paper addresses the sampling of molecular conformations at equilibrium, and notably in a computationally efficient manner that does not require repeated sampling (e.g. MCMC or running molecular dynamics). The authors propose a normalizing flow generative architecture, building from the recent TarFlow, that notably now operates at a scale and performance that generalizes to different molecules and systems. Evaluations are conducted on peptide systems up to octapeptides.

**Questions:**

1. What is the impact of each of the proposed modifications for the resulting architecture? Which components had the most impact for the empirical results?
2. What practical obstacles arise from either extending to or evaluating on systems beyond octopeptides?

**Ethical Concerns:**

["NO or VERY MINOR ethics concerns only"]

**Final Justification:**

While empirical in nature, the paper does present a notable empirical jump in learning-based conformation sampling, and the authors have done a good job with comparisons and ablations for future readers to understand the true role of their work.

**Limitations:**

yes

**Quality:**

3

**Strengths And Weaknesses:**

Writing and presentation in general is quite clear, and the order is logical. The empirical work is strong: improvements from the relevant baselines are notable, multiple sampling schemes are analyzed. Metrics for results are modern and appropriate; compute budgets and hyper-parameters are documented and reproducible. Generalization on up to length 8 peptide systems is also significant.

The primary concern for the work is academic novelty compared to previous methods like TarFlow. More detailed experimental analysis on all of the modifications in Section 3.1 and each of their roles would help further readers to understand the roles of each individual component of the proposed architecture.

---

> ### Author Rebuttal · Authors · 2025-07-31
>
> # Response to Reviewer EUPF
>
> We thank the reviewer for their time and effort in reviewing our work. We are glad that the reviewer describes the scale and generalization to other molecular systems as "notable", the empirical evaluation as "strong" and "significant", and the presentation as "clear". In what follows, we address the concerns raised and provide suggested evaluations.
>
> > More detailed experimental analysis on all of the modifications in Section 3.1 and each of their roles would help further readers to understand the roles of each individual component of the proposed architecture. What is the impact of each of the proposed modifications for the resulting architecture? Which components had the most impact for the empirical results?
>
> In Section 4.3, we describe the ablation study of all the architectural changes introduced in our work. In particular, Table 3 compares the results of the original TarFlow architecture with the proposed architecture that includes the "adapt + transition" blocks as well as the proposed chemistry-aware permutations. We observe that both changes provide a performance boost; notable in particular given the chemistry-aware permutations imply no increase in training or inference wall-time. The walltime increase of "adapt + transition" is only pronounced on the smaller systems and is not significant on the 8AA dataset (Figure 22, supplementary material).
>
> > What practical obstacles arise from either extending to or evaluating on systems beyond octopeptides?
>
> Following the reviewer's suggestion, in the table below, we report the performance of our method on the 9 residue peptide introduced by [1] and the well-studied small protein Chignolin. The reported metrics demonstrate that the proposed method outperforms all of the baselines. Note that the training data only contains systems of length up to 8 amino acids, which suggests that the method generalizes across systems of previously unseen length.
>
> We further explore an additional step in the self-refinement procedure for Chignolin, where an initial set of 1000 samples from Ensemble undergo constrained energy minimization with restraints on the macrostructure. The resulting configurations are then used to finetune our model before running the self-refinement training.
>
> |  | YQNPDGSQA (9AA) |  |  |  | Chignolin (10AA) |  |  |  |
> | :- | - | :- | :- | :- | - | :- | :- | :- |
> |  | ESS | E-W2 | Torus-W2 | TICA-W2 | ESS | E-W2 | Torus-W2 | TICA-W2 |
> | Ensemble (SNIS) | 0.0062 | 24.78 | 3.73 | 1.39 | 0.0001 | 829.59 | 4.21 | 0.73 |
> | Ensemble (self-refine + SNIS) | 0.0082 | 21.34 | 3.77 | 1.39 | 0.0003 | 447.72 | 4.64 | 1.002 |
> | Ensemble + constrained energy min + self-refine | \- | \- | \- | \- | 0.0002 | 407.13 | 4.52 | 0.81 |
> | BioEmu | - | 1629.55 | 3.85 | 1.03 | - | 9.74e9 | 3.74 | 0.89 |
> | UniSim | - | 1630.42 | 5.65 | 1.05 | - | 3.21e9 | 6.06 | 1.26 |
>
> We thank the reviewer again for the detailed feedback and constructive suggestions on the improvement of the manuscript. We hope we address their main concerns, and we remain available throughout the discussion period for further clarifications and suggestions for improvement.
>
> [1] Xiongwu Wu and Bernard R Brooks. "Beta-hairpin folding mechanism of a nine-residue peptide revealed from molecular dynamics simulations in explicit water." Biophys J. (2004)

---

> > ### Author Response · Authors · 2025-08-07
> >
> > We once again thank the reviewer for their time invested in considering our work. We hope our previous response, including the clarification of ablation experiments and the inclusion of larger 9AA and 10AA evaluation systems has strengthened the work for the reviewer and addressed their concerns raised. As the discussion period is coming to a close, we politely remind the reviewer that we remain available to address any final questions they may have.

---

> > > ### Comment · Reviewer_EUPF · 2025-08-07
> > >
> > > I appreciate the authors' responses to my questions and the additional experiments. I have no further questions and will keep my score.

---

### Official Review · Reviewer_VbSw · 2025-07-03

**Clarity:** 3
**Significance:** 3
**Originality:** 2
**Rating:** 5
**Confidence:** 2

**Summary:**

This paper addresses the challenge of efficiently sampling molecular conformations. It introduces ENSEMBLE, a large-scale, all-atom transferable normalizing flow model trained on molecular dynamics data of peptides ranging from 2 to 8 residues. ENSEMBLE enables fast, zero-shot sampling across different peptide sequences and lengths, surpassing traditional methods. ENSEMBLE is demonstrated through extensive experiments and various systems, while the trajectory data is also provided.

**Questions:**

Most of my questions reading the papers has been resolved in the supplementary tool, I thank the authors for the detailed appendix.

**Ethical Concerns:**

["NO or VERY MINOR ethics concerns only"]

**Final Justification:**

My minor concerns have been resolved, and I keep my score accordingly.

**Limitations:**

yes,  the authors adequately addressed the limitations and potential negative societal impact of their work

**Quality:**

2

**Strengths And Weaknesses:**

**Strengths**

1. **(Significance) Scalable and transferable approach**

    Unlike prior works that focus on small systems [1] or transferability within fixed-length peptides like dipeptides [2], the proposed method scales up to peptides of up to 8 residues and demonstrates transferability across varying lengths. This represents a meaningful step forward in general-purpose, all-atom peptide modeling.

2. **(Clarity) Clear and well-organized presentation**

    The paper is well structured and easy to follow. The inclusion of comprehensive experiments across multiple peptide systems, as well as ablation studies, effectively supports the authors’ claims.


**Weaknesses**

1. **(Quality, minor) Performance on 2-residue (2AA) systems**

    As acknowledged by the authors, the method underperforms on 2AA systems compared to other lengths. While this is a minor issue and does not detract significantly from the overall contributions, it may indicate areas where the model’s generalization could be improved.

2. **(Quality, minor) Efficiency claim could be better quantified**

    While the method outperforms baselines under fixed energy evaluation budgets, adding a simple experiment showing performance as a function of energy evaluations (e.g., performance-vs-budget curves on one system) would strengthen the claim of efficiency and allow for better interpretability of trade-offs.

    **Minor issues**

- There is a missing parenthesis “(” after “4AA” in the first line of Table 3.
- Bold formatting appears to be missing for “TarFlow base” and “backbone permutation” in Table 3, which may reduce clarity.

---

> ### Author Rebuttal · Authors · 2025-07-31
>
> # Response to Reviewer VbSw
>
> We thank the reviewer for their detailed feedback and their time spent. We appreciate their positive evaluation, in particular, the reviewer describes the proposed approach as "scalable and transferable", the empirical study as "extensive", and the presentation as "clear and well-organized". In what follows, we provide requested clarifications.
>
> > (Quality, minor) Performance on 2-residue (2AA) systems. As acknowledged by the authors, the method underperforms on 2AA systems compared to other lengths. While this is a minor issue and does not detract significantly from the overall contributions, it may indicate areas where the model's generalization could be improved.
>
> The fact that Ensemble slightly underperforms on 2-residue systems is the consequence of ECNF being trained only on 2 residues. Indeed, ECNF++ -- our attempt to scale this approach to 4 residues (following the improved training techniques from [1]), demonstrates worse performance across both 2 and 4 residues.
>
> > (Quality, minor) Efficiency claim could be better quantified. While the method outperforms baselines under fixed energy evaluation budgets, adding a simple experiment showing performance as a function of energy evaluations (e.g., performance-vs-budget curves on one system) would strengthen the claim of efficiency and allow for better interpretability of trade-offs
>
> In Fig. 1 (page 2), we report the performance (across 30 unseen 4-residue systems) as a function of energy evaluations and GPU walltime. Notably, the proposed method outperforms Molecular Dynamics (MD) simulations for the same evaluation budget, as the latter fails to capture all metastable states (see the 3rd plot in the top row).
>
> > Minor issues
>
> We thank the reviewer for a thorough reading of the manuscript. We will fix those typos.
>
> We thank the reviewer again for the detailed feedback and constructive suggestions on the improvement of the manuscript. We hope we address their main concerns, and we remain available throughout the discussion period for further clarifications and suggestions for improvement.
>
> [1] Tan, Charlie B., Avishek Joey Bose, Chen Lin, Leon Klein, Michael M. Bronstein, and Alexander Tong. "Scalable equilibrium sampling with sequential boltzmann generators." arXiv preprint arXiv:2502.18462 (2025).**

---

> > ### Comment · Reviewer_VbSw · 2025-08-06
> >
> > My minor concerns have been resolved, and I keep my score. I thank the authors for the clarification.

---

> > > ### Author Response · Authors · 2025-08-07
> > >
> > > We thank the reviewer for their response, and are pleased that their “minor concerns have been resolved”. We once again thank the reviewer for their time and effort reviewing our work.

---

### Note · Authors · 2025-08-16

Dear Area Chair and Esteemed Reviewers,

As the discussion period concludes, we wish to express our sincere gratitude for the thoughtful and constructive engagement from all reviewers. We are encouraged that our clarifications, new experiments, and additional baselines have resolved initial concerns and led to improved assessments of our work. We summarize below how these points were addressed, leading to a consensus in favor of our work.

* Reviewer VbSw (Maintained score 5): The reviewer raised only minor concerns and questions regarding the performance of our method for 2AA and fixed energy budget evaluations. We addressed these concerns during the discussion, to which the reviewer confirmed their concerns were resolved.
* Reviewer EUPF (Maintained score 4): We clarified the architectural modifications over TarFlow and reported new results on 9- and 10-residue peptides, showing superior performance to baselines like UniSim and BioEmu. The reviewer acknowledged these additions and maintained their score.
* Reviewer 7hoH (Increased score from 3): The reviewer raised concerns regarding extended MD baselines, additional metrics (e.g., JSD), and evaluations on larger peptides. In response, we clarified a misunderstanding around the apparent short length-scale of the baseline MD simulations, noting that this was due to adhering to a fixed energy budget for fair comparison, but also provided longer length-scales to compare against our method. We also reported the requested additional metrics and presented new evaluations on larger peptides, demonstrating strong performance against established baselines. The reviewer acknowledged the thoroughness of our response and raised their score.
* Reviewer qKai (Increased score from 2): We engaged in detailed discussions around novelty, architectural choices (transformer vs GNNs), method clarifications, and concerns over 2AA performance. Additionally, following the reviewer’s suggestions, we report ESS/s metric, additional baselines i.e., UniSim, BioEmu, and Timewarp, and report the ESS/s metric. The reviewer confirmed their concerns were resolved and raised their score.

We have provided comprehensive empirical and theoretical evidence to address the reviewer's critique at every stage. We believe these additions strengthen our work as a new benchmark for amortized sampling. We thank the reviewers and AC for their time, and hope the Area Chair will consider the entirety of this discussion in their final assessment.

---

### Decision · Program_Chairs · 2025-09-17

**Decision:**

Accept (poster)

**Comment:**

The paper introduces a normalizing flow framework for sampling conformers, with the aim of improving generalization across molecules. This is an important problem and the present approach is well-grounded, and obtains good empirical results on a peptide dataset. The consensus from reviewers was generally positive reviewers highlighting both the potential impact of improved sampling methods and the practicality of the proposed framework. While some weaknesses about novelty and experimental validation were raised, the positives outweigh the negatives, therefore I recommend acceptance.